

# Research on the Extraction of Pre-Seismic Anomalies in Borehole Strain Data of the Mado Earthquake Based on the SVMD-Informer Model

Shanzhi Dong [1, 2], Jie Zhang [1, 2], Changfeng Qin [1, 2], Yu Duan [1, 2], Chenyang Li [1, 2], Chengquan Chi[1, 2, *],
5  Zhichao Zhang[1, 2, *]

[1] School of Information Science and Technology, Hainan Normal University, Haikou, 571158, China
[2] Key Laboratory of Data Science and Smart Education, Hainan Normal University, Ministry of Education, Haikou, China

*Correspondence to*: Chengquan Chi, Zhichao Zhang (chicqhainnu@gmail.com, zhangzhichao@hainnu.edu.cn)

**Abstract.** Earthquake is a major natural disaster triggered by the accumulation and release of crustal stress, and the accurate
10  extraction of pre-seismic anomaly signals is crucial to improve the earthquake prediction capability. In this study, an
anomaly detection method for borehole strain data based on the combination of Segmented Variational Modal
Decomposition (SVMD) and Informer network is proposed, and a pre-seismic anomaly extraction study is carried out for the
2021 Mado Ms7.4 earthquake in Qinghai. The SVMD method effectively solves the memory limitation problem of
traditional Variational Modal Decomposition (VMD) when dealing with large-scale data through the sliding-window
15  mechanism, and at the same time maintains the correlation between the data. The Informer network significantly reduces the
computational complexity of the long-series prediction and realizes the high-precision one-time long-time series prediction
by utilizing its *ProbSparse* self-attention mechanism and self-attention distillation. By analyzing the borehole strain data
from the Mengyuan station, this study identifies the accelerated anomaly accumulation phenomenon in the two stages before
the Mado earthquake: in the first stage, the number of anomalous days shows an accelerated growth starting from about three
20  months before the earthquake (February 13, 2021); in the second stage, the anomalous accumulation tendency is further
intensified since the second month before the earthquake (the end of March, 2021), and the accumulation curve shows a
typical S-shape growth characteristic. The results are highly consistent with the time windows of the index of microwave
radiation anomaly (IMRA), outward long-wave radiation (OLR) and geoelectric field anomalies, and with the subsurface-to-
atmosphere multilayer anomalies (e.g., Benionff strain, CO concentration, electron concentration anomalies, etc.), which
indicate that the borehole strain anomalies are closely related to the gestation process of the Mado earthquake. This study
provides a new method for the extraction of pre-seismic anomalies based on machine learning, and provides an important
basis for understanding earthquake precursors.



## 1 Introduction

Earthquakes are caused by the release of accumulated stresses in the Earth's crust beyond the strength of the rocks during
plate interactions and collisions. Once the stress in the earth's crust exceeds a certain critical threshold, the crust will rupture,
releasing seismic waves reflected through the ground, which can cause great damage (Kanamori and Brodsky, 2001; Fan et
al., 2021). Earthquakes can damage infrastructure such as the ground, transportation, and buildings, and may trigger
secondary disasters such as volcanic eruptions, tsunamis, and landslides, which have serious impacts on the quality of life,
development, and civilization of human societies (Peptan et al., 2023; Fan et al., 2019; Potter et al., 2015). Studies have
shown that more than 80% of earthquakes of magnitude 5.0 and above in mainland China in 2022 occur within the annual
seismic hazard zone  (Yu et al., 2024a). Therefore, it is crucial to study the anomalies that precede earthquakes.

The study of earthquake precursor data is important for deepening our understanding of seismic activity and its potential
impacts. By deeply analyzing patterns and trends in the data, researchers are able to identify early signs that may signal an
impending earthquake. As data analysis techniques continue to evolve, predictive models can be updated and optimized,
which not only facilitates collaboration among scientists, but also contributes to the assessment and management of seismic
hazards globally, thereby improving the ability to respond to them. Researchers around the world have explored a wide range
of phenomena before and after earthquakes, covering different structural levels of the Earth, including the subsurface,
surface, and atmospheric domains. (Ma et al., 2022) identified geodetic anomalies prior to the Mado earthquake through
observed GPS values and computationally analyzed b-values, and verified that the pre-seismic anomalies of GPS data, b-
values, and stress and strain accumulations were associated with the Mado earthquake. (Liu et al., 2023) used day-by-day
NOAA satellite long-wave radiation (OLR) data to investigate the changes in the OLR anomalies prior to the 2023 Turkey
magnitude 7.8 earthquake, and found that the OLR anomalies were synchronized with the cycle of tidal stress change and
mainly concentrated in the southwestern section of the East Anatolian Fracture Zone. The study suggests that the emergence
of OLR anomalies may be related to the accumulation of tectonic stresses to a critical state during the earthquake gestation
process, which reflects the thermal infrared signals of the stress changes prior to the occurrence of earthquakes. (Guo et al.,
2015) based on the GPS data estimated the of global total electron content (TEC), analyzed ionospheric disturbances that
may be related to earthquakes, and found significant TEC anomalies prior to the April 11, 2012 magnitude 8.6 Sumatra
earthquake and magnitude 6.7 Mexico. In addition, other scholars have investigated other parametric indicators and fields
such as subsurface fluids (Chen et al., 2013), gravity field (Yiqing et al., 2011), geomagnetism (Li et al., 2019), microwave
bright temperature (MBT) (Qi et al., 2021; Qi et al., 2022), CO (Cui et al., 2024), and electron density (Han et al., 2023). All
these studies provide abundant data support and theoretical basis for the exploration of earthquake precursors, and provide
valuable references for our understanding of the potential mechanisms of earthquake occurrence and prevention of seismic
hazards.

Since the implementation of the U.S. Earth Lens Program, a large amount of open data has been accessed and utilized,
and the Plate Boundary Observations (PBO) program has contributed to the development of strain techniques. Borehole



strain observations have received unprecedented attention because of their high resolution and sensitivity (Lou and Tian 2022; Roeloffs, 2010; Barbour and Agnew, 2012). Borehole strain observations are an important tool for studying crustal deformation and changes in the ground stress field. Under the action of regional stress field, crustal deformation will show subtle changes. By mounting strain gauges deeper in the bedrock, borehole strain gauges are able to continuously record

stress and strain data, making them a key tool for monitoring crustal deformation. The high resolution recordings provided by borehole strain gauges allow us to capture small changes in strain, thus providing accurate data to support a deeper understanding of crustal deformation processes. Borehole strain observations are superior to GPS and laser strain gauges in capturing short- and medium-term strain changes and pre-seismic strain changes (Qiu and Shi, 2004). Four-component borehole strain observation is an important geodynamic observation technique independently developed in China. In the

digital seismic observation network in China, the YRY-4 four-component strain gauge is used as the main observation equipment, which not only provides the four-component data, but also records auxiliary observations such as solid tides, air temperature, and air pressure (Chi, 2009; Tang et al., 2023b). In addition to the application in earthquake precursor research, the borehole strain observation data also play an irreplaceable role in the research fields of slow earthquakes, co-seismic stress triggering, tremor, earth free oscillation, and seismic wave propagation, and many unique results have been achieved

(Qiu, 2014).

Numerous scholars have accumulated rich research experience and results in extracting and identifying pre-seismic anomalous signals using borehole strain observation data. (Chi, 2013) inferred that these anomalies were closely related to the strain precursors in the preparation of two earthquakes by analyzing the tidal anomalies recorded by borehole strain gauges during the preparation of Wenchuan and Lushan strong earthquakes. (Zhu et al., 2020) observed the Wenchuan

earthquake precursors by analyzing the eigenvalues and eigenvectors of the borehole strain data. (Yu et al., 2020) constructed a complex network using multi-channel singular spectrum analysis (MSSA) using borehole strain data from the southwestern terminus of the Sichuan-Yunnan Longmenshan rift zone, and the results showed that the network provides a powerful tool for earthquake precursor monitoring. (Li et al., 2024a) successfully extracted the pre-seismic anomalies of the Jiuzhaigou 7.0 magnitude earthquake by utilizing variational modal decomposition (VMD) to preprocess the borehole strain

observation data and combining it with the Graph WaveNet (GWN) model for multi-station data. (Wu, 2012) revealed the intrinsic evolutionary characteristics of the seismic process through quantitative simulation of cluster sub-statistical equations and established a link between them and the dynamic change of earthquake precursors. (Liu et al., 2014)Liu et al. (2014) analyzed the time-frequency characteristics of borehole strain data using the S-transform method and revealed reliable anomalies reflecting the whole process of strain changes before and after the Lushan earthquake. In addition, (Chi et

al., 2019) extracted the Wenchuan earthquake borehole strain data anomalies by decomposing the borehole strain signal into multiple modes using VMD and using a new state-space model to determine the number of decomposed modes, and then calculating eigenvalues used to detect anomalies associated with the earthquake by using the anomaly detection method of principal component analysis (PCA). These studies show that borehole strain observations have significant advantages in precursor anomaly extraction and seismic correlation analysis.





Borehole strain observation can capture the subtle phenomena during seismic activities in a timely manner, and its observation data can reflect the stress-strain changes in rocks, thus providing a potential method for extracting strain anomalies before earthquakes. However, due to the high precision and wide bandwidth characteristics of borehole strain observation, it is highly susceptible to interference from external environmental factors. To address this problem, researchers have conducted in-depth studies on how to remove external interference from borehole strain data. (Qiu et al., 2011)

extracted pre-earthquake borehole strain anomalies by using high-pass filtering to remove the interference from seasonal variations and long-period signals. Subsequently, (Zhang et al., 2019) performed first-order difference processing on the borehole strain data to significantly attenuate the low-frequency effects such as solar and lunar gravitational tidal forces in the measured data, and meanwhile, wavelet transform based on the first-order difference data enhanced the short-periodic component of the strain signal. In recent studies, (Zhu et al., 2020), (Yu et al., 2021), and (Li et al., 2024a) removed the

strain response due to seasonal trends, barometric pressure, solid tides, and water level variations based on harmonic analysis, state-space modeling, and variational modal decomposition (VMD) methods, respectively, to isolate the non-constructive perturbations effectively. In addition, (Zhu et al., 2024) firstly removed the long-term background trend and cyclic trend of the borehole strain observation data before the Lushan earthquake by time-series decomposition, secondly, decomposed the data by multi-channel singular spectrum analysis to eliminate the strain response due to water level and air pressure, and

finally extracted the pre-earthquake negentropic anomaly in the borehole strain data.

        VMD (Variational Modal Decomposition) transforms the signal decomposition into a variational modal decomposition process, which realizes the adaptive segmentation of each component in the frequency domain. The method is especially suitable for analyzing nonlinear and nonsmooth signals such as step, jump, and burr. In recent years, VMD has been widely used in earth sciences and related fields, and its processing of seismic signals is significantly better than EMD (Empirical

Modal Decomposition) and its derivative methods (Rao et al., 2024; Li et al., 2018; Xue et al., 2019). However, with the continuous accumulation of seismic precursor observation data and its increasing complexity, the traditional signal processing techniques gradually show their limitations. Due to the variation of data frequency and dynamic displacement, it is often difficult for traditional filters to accurately capture key information, which may even lead to missing or blurred information. Although the variational modal decomposition (VMD) technique performs well in the field of data

decomposition, it often faces the problem of memory overflow when dealing with large-scale datasets. For this reason, this paper adopts the SVMD method, which effectively solves the memory limitation problem through the sliding window mechanism while ensuring the correlation between the data. The studies of (Chi et al., 2023) and (Li et al., 2024b) have shown that SVMD outperforms the traditional VMD method in terms of computational efficiency and memory utilization when dealing with large-scale data.

In recent years, machine learning techniques have become promising tools for studying earthquake precursor data. Studies have been conducted using random forest (RF) models (Akhoondzadeh, 2009; Asim et al., 2016; Tsuchiya et al., 2024), decision trees (Sikder and Munakata, 2009), GRU (Chi et al., 2023), LSTM (Wang et al., 2020; Zhang and Wang, 2023, 2024), neural networks (Kail et al., 2022; Yousefzadeh et al., 2021; Bilal et al., 2022), and other methods that show



significant potential in the prediction of seismic events, for example, estimating earthquake location and magnitude, while
reducing the false alarm rate. Despite the remarkable progress, challenges remain in terms of data quality, scarcity, and
heterogeneity. With the continuous development and optimization of machine learning models and the continued availability
of high-quality data, the accuracy and reliability of earthquake prediction is expected to be further improved.

In this paper, we propose an innovative method based on SVMD-Informer network for extracting pre-seismic anomaly
information from borehole strain data, and take the Mado earthquake as an example to analyze the borehole strain data of
Mengyuan Terrace. The SVMD-Informer network used in this paper is not only good at capturing the features of the
borehole strain data, but also demonstrates high prediction accuracy. The prediction intervals are constructed by statistical
methods, thus realizing efficient anomaly detection. The research in this paper can provide more accurate pre-earthquake
anomaly information for the earthquake early warning system and improve the accuracy and timeliness of earthquake
prediction.

In order to analyze the correlation between the borehole strain data from Mengyuan station and the Mado earthquake, the
process shown in Fig. 1 was used in this study. As shown in Fig. 1, first, the four-component borehole data were converted
to strain data. Subsequently, the borehole strain data were decomposed using the Segmented Variational Modal
Decomposition (SVMD) method, and the decomposition results were fused to remove the interference of influencing factors
such as the annual trend and the solid tidal response. The fused data were used as inputs for training, validation and
prediction using the Informer model. In the prediction stage, prediction intervals for the upper and lower bounds are
established based on the model output, and anomalous signals are identified by comparing the prediction intervals with the
original data. Finally, the pre-seismic anomalous cumulative values in the borehole strain data from Mengyuan station were
statistically analyzed, and the significant anomalous signals were successfully extracted.

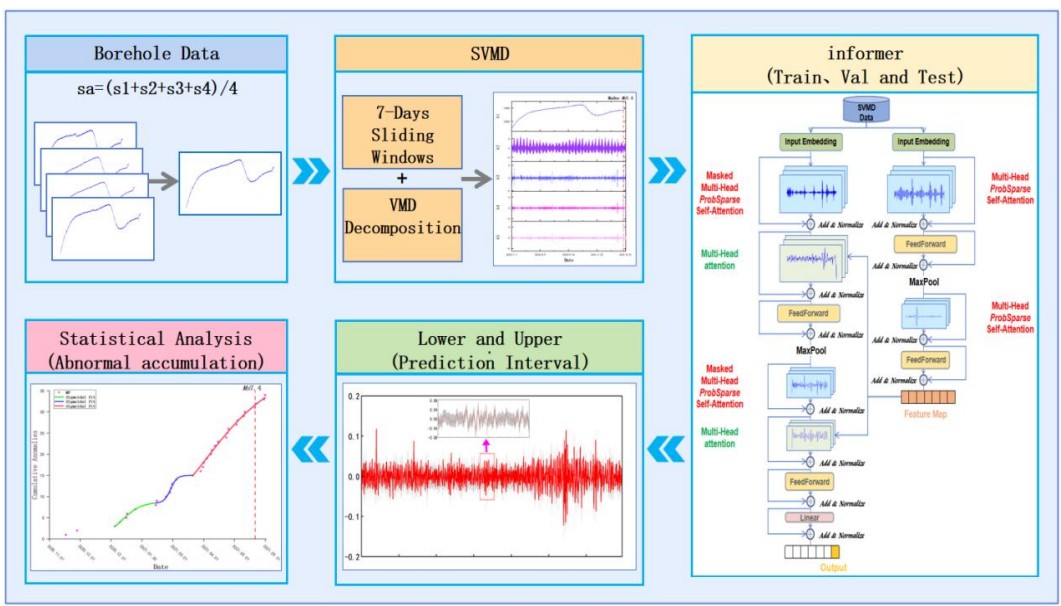

**Figure 1. Framework for borehole strain data processing and pre-seismic anomaly extraction.**

Ⓒ Author(s) 2025. CC BY 4.0 License.



## 2 Observational data and earthquakes

### 2.1 Borehole strain data

In recent years, several studies have verified the reliability of the sampled data from high-component borehole strain gauges,
demonstrating the ability of four-component borehole strain gauges to record strained seismic waves from earthquakes of
different magnitudes (Tang et al., 2023a; Qiu et al., 2015). As an ideal tool for crustal motion observation, the observation
technology of the four-component borehole strain gauge is gradually maturing, which provides important support for the
identification of earthquake precursors (Qiu et al., 2021). The YRY-4 four-component borehole strain gauge is one of the
core instruments for the observation of crustal deformation in China, and it has the significant advantages of high sensitivity,
wide observation band, good data consistency, and superior long-term stability. Borehole Strain Gauge is to place the sensor
in the borehole for observation, relative to the earth its observation is an extremely small part of the crust deformation, can
be approximated as a point of deformation observation results. The four-component borehole strain observation represents a
relative observation that is capable of detecting changes in the target observation, but does not provide a complete
measurement of the target observation. Its characterization is determined by the underlying principles of its model. The four
probes on the borehole strain gauge are evenly spaced at 45° intervals to ensure multi-directional monitoring of crustal
deformation. The measured value of any one element is recorded as $S_1$, rotated by 45° in turn, and the remaining three
elements are recorded as $S_2$, $S_3$, and $S_4$. The amount of change in the observed values of the four elements satisfies the self-
consistent equation (1):

$$S_1 + S_3 = k\left(S_2 + S_4\right) \tag{1}$$

This equation can be used to estimate the reliability of the data. $k$ is the self-consistency coefficient, and in the ideal case $k$
= 1. We consider the data to be reliable when $k \geq 0.95$. Under plane strain conditions at or near the Earth's surface, only
three independent variables are considered. Therefore, we can derive the various strains from the Mengyuan station record
using equation (2) as follows:

$$\begin{cases} S_{13} = S_1 - S_3 \\ S_{24} = S_2 - S_4 \\ S_a = \left(S_1 + S_2 + S_3 + S_4\right)/2 \end{cases} \tag{2}$$

All three substitutions are important. Among them, $S_{13}$ and $S_{24}$ represent shear strains independent of each other, and $S_a$ is
the surface strain. Compared with the shear strains $S_{13}$ and $S_{24}$, the surface strain $S_a$ is more representative of the four
components measured by the YRY-4 borehole strain gauge, and therefore the data characterization of the surface strain $S_a$ is
taken as the object of study in this paper.





## 2.2 Maduo earthquake

According to the China Earthquake Networks Center (CENC), the Ms7.4 magnitude earthquake occurred at 02:04:03 on
May 22, 2021, in Mado County, Qinghai Province, with an epicenter located at 98.34 °E, 34.59 °N, about 70-80 km from the
East Kunlun Fracture Zone, at a depth of 17 km (Zhu et al., 2021). This great earthquake is located in the Bayankala block,
which is one of the main blocks of the Qinghai Tibet Plateau orogeny and one of the regions with the most frequent seismic
activity in China (LÜ et al., 2022). The sudden-onset earthquake was the largest earthquake to occur in China since the 2008
Wenchuan Ms7.9 earthquake. As of 8 a.m. nine days after the earthquake, the China Earthquake Administration recorded a
total of 2,979 aftershocks, including one Ms5.1, 13 Ms4.0 ~ 4.9, and 63 Ms3.0 ~ 3.9, which caused severe damage to
buildings and roads in the area (Wang et al., 2021).

Dobrowolski's estimate of the radius of influence of precursors for earthquakes of different magnitudes is shown in
equation (3):

$$\rho = 10^{0.43M} \, km \tag{3}$$

where $M$ denotes the magnitude and $\rho$ denotes the radius of influence of an $M$ magnitude earthquake. The radius of
influence of the Mado earthquake is about 1377.2km. In order to analyze the seismic effects near the epicenter, we selected
the Menyuan station, which is closer to the epicenter of the Mado earthquake, with a distance of 422.06 km from the
epicenter. By analyzing the data collected at Menyuan station, we extracted 34 anomalous signals. This result indicates that
the Menyuan station has the potential and ability to monitor earthquake-related anomalies. Figure 2 visually depicts the
geographic location of the Menyuan station relative to the epicenter of the Mado earthquake.

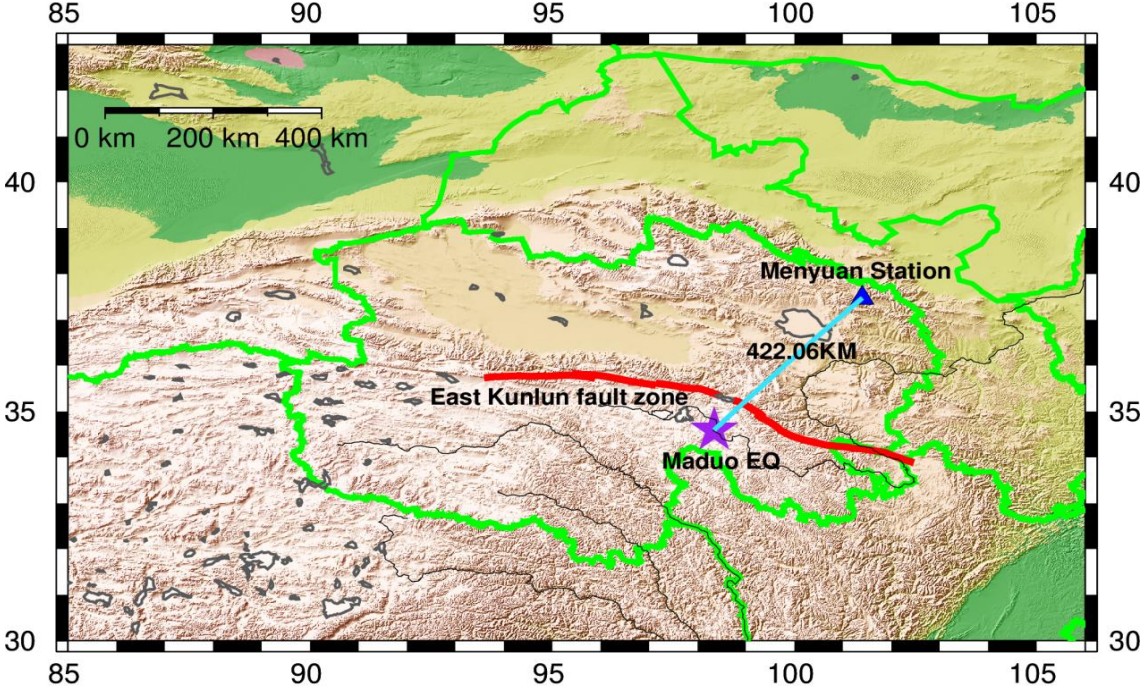



**Figure 2. Location of the Mengyuan observatory relative to the epicenter of the Mado earthquake. The blue triangle represents the location of the borehole strain observatory. The purple star represents the location of the epicenter of the Mado earthquake, while the red line inscribes the East Kunlun Fault Zone. This map was generated by GMT software, v. 6.0.0rc5 (https://gmt-china.org/).**

## 3 Method

### 3.1 Segmented variational modal decomposition (SVMD)

Since the introduction of VMD by (Dragomiretskiy and Zosso, 2014), it has achieved better results in dealing with data nonlinear, nonsmooth, and nonsmoothed signals. VMD is a fully implicit, adaptive, and completely nonrecursive approach to modal variational and signal processing. VMD is developed based on the classical Wiener filter denoising and Fourier transform, which aims to decompose the time series data into a series of intrinsic modal functions (IMFs) with finite bandwidth. The decomposition process essentially involves solving the variational problem, and the variational model can be expressed as follows:

$$\min_{\{u_k\},\{\omega_k\}} \sum_{k=1}^{K} \left\| \partial_t \left\{ \left[ \left( \delta(t) + \frac{j}{\pi t} \right) * u_k(t) \right] e^{-j\omega_k t} \right\} \right\|_2^2$$

$$st \sum_{k=1}^{K} u_k(t) = f(t) \tag{4}$$

In equation (4) $\{u_k\} := \{u_1, u_2, ..., u_K\}$ and $\{\omega_k\} := \{\omega_1, \omega_2, ..., \omega_K\}$ are the corresponding center frequencies of the $k$ modal functions and signal decompositions, respectively. Similarly, $\Sigma_k := \sum_{k=1}^{K}$ is a shorthand notation for all modes and their center frequencies, respectively.

The quadratic penalty term $\alpha$ and the Lagrange multiplier $\lambda(t)$ are introduced in the VMD in order to solve the constraint problem, which benefits from both the good convergence properties of the finite power quadratic penalty and the strict enforcement of the constraint by the Lagrange multiplier. Thus, the augmented Lagrangian is introduced:

$$L\left(\{u_k\}, \{\omega_k\}, \lambda\right) := \alpha \sum_k \left\| \partial_t \left[ \left( \delta(t) + \frac{j}{\pi t} \right) * u_k(t) \right] e^{-j\omega_k t} \right\|_2^2$$

$$+ \left\| f(t) - \sum_k u_k(t) \right\|_2^2 + \left\langle \lambda(t), f(t) - \sum_k u_k(t) \right\rangle \tag{5}$$

In the equation $L$ denotes the Lagrange promotion operator, $\alpha$ denotes the data fidelity constraint function, and $\lambda$ denotes the Lagrange multiplier. Meanwhile the alternating direction multiplier method (ADMM) is used in the VMD to find $u_k$, $\omega_k$, $\lambda$ and to solve the updated iterative optimization of equation (15). Where the above equations get the solution formula of the model as $u_k$:

$$\hat{u}_k^{n+1}(\omega) = \frac{\hat{f}(\omega) - \sum_{i \neq k} \hat{u}_i(\omega) + \frac{\hat{\lambda}(\omega)}{2}}{1 + 2\alpha(\omega - \omega_k)^2} \tag{6}$$



The equation $\omega_k$ for the center frequency is:

$$\omega_k^{n+1} = \frac{\int_0^\infty \omega |\hat{u}_k(\omega)|^2 \, d\omega}{\int_0^\infty |\hat{u}_k(\omega)|^2 \, d\omega} \tag{7}$$

The Lagrange multiplier $\lambda$ is:

$$\hat{\lambda}^{n+1}(\omega) \leftarrow \hat{\lambda}^n(\omega) + \tau\left(\hat{f}(\omega) - \sum_k \hat{u}_k^{n+1}(\omega)\right) \tag{8}$$

Although VMD has the advantages of good performance and greater robustness to sampling and noise relative to existing mode decomposition models, when faced with a large number of data computations, problems such as slow data processing and computer memory limitations may arise because the VMD method requires a global search and solves the variational segmentation problem. Therefore, in this paper, we adopt a method that applies variational modal decomposition (VMD) to data segments by combining sliding windows (SVMD). This method not only effectively solves the above problems in VMD.

And through the sliding window mechanism, the correlation between data points can be effectively maintained (Chi et al., 2023).

The principle of SVMD method is shown in Fig. 3. We choose the sliding window mechanism with a size of 7 days and a sliding step of 1 day to realize the data segmentation. First we set the initial window as all the data from the first day to the seventh day and perform VMD decomposition of the data within that window. Starting from the second sliding window,

only the results of the VMD decomposition of the current window are retained and superimposed with the decomposition results of the previous window, and in this logical order, the data are processed sequentially, and finally the complete dataset processed by SVMD is obtained.



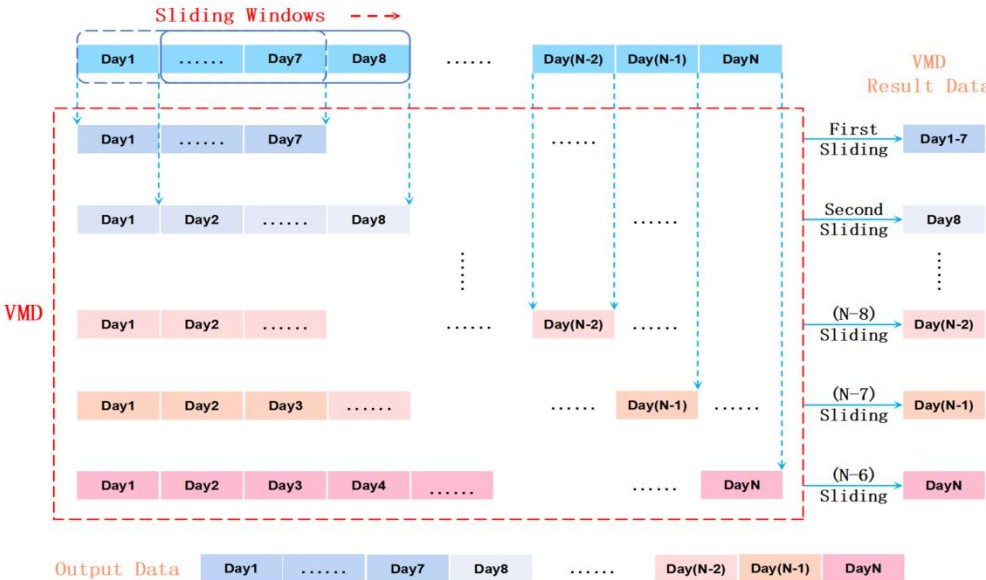

**Figure 3. Schematic diagram of the Segmented Variational Modal Decomposition (SVMD).**

## 3.2 Informer network

In recent years, deep learning techniques have been widely used in the field of seismic hazard recognition and prediction, injecting new vigor into the development of earthquake science. Although traditional sequence models, such as Long Short-Term Memory (LSTM) networks, Gated Recurrent Units (GRU), and Recurrent Neural Networks (RNN), have made significant contributions to time series prediction, they have obvious limitations in capturing long-range dependencies when dealing with the task of Long Sequence Time-series Forecasting (LSTF). In contrast, the Transformer model, with its self-attentive mechanism, is able to reduce the maximum path length of network signals to a theoretical minimum while avoiding the complexity of the loop structure, thus showing great potential for LSTF problems (Ashish et al., 2017). However, there are some serious issues with the traditional Transformer model that hinder its application to LSTF, including quadratic time complexity, high memory usage, inherent limitations of the encoder-decoder architecture, and the drawback of needing to rely on the previous prediction result for the current prediction. To address these issues, this paper employs the Informer model, which significantly improves the performance of long-series time series prediction while maintaining efficient computation.

Since the Informer model was proposed by (Zhou et al., 2021), it has been widely used in the fields of river runoff time series prediction (Tepetidis et al., 2024), long-term prediction of indoor air quality (Long et al., 2023), wind power series prediction (Wei et al., 2023), financial time series prediction (Zhang et al., 2024), automatic driving trajectory prediction (Chen et al., 2023), etc. These applications fully show that Informer model has significant prediction performance



advantages. The efficiency and accuracy of Informer model provide innovative ideas for earthquake research, especially in the analysis and prediction of earthquake precursor data, which shows great potential.

The Informer model has three significant features:

(1) Adoption of *ProbSparse* (i.e. Probabilistic Sparsity) self-attention mechanism, which reduces the time complexity and memory occupation from $O(L^2)$ of Transformer model to $O(L \log L)$ .

(2) Self-attention Distilling is proposed for downsampling operation to reduce the parameters of dimensionality and network parameters.

(3) Generation Style Decoder is utilized, which can effectively predict long time sequences at one time instead of step
by step, improving the inference speed of prediction.

The Informer model consists of three main components: encoder layer, decoder layer and prediction layer. Compared to the traditional self-attention methods, the encoder mainly deals with longer sequence inputs by using sparse self-attention. Self-attention Distilling refers to the distillation of the extracted self-attention operation, which significantly reduces the size of the network.Multi-Head *ProbSparse* self-attention denotes the self-attention extracted block, which is used to stitch
together the feature maps. Finally, the decoder processes the inputs from the long time series and zeroes out the target elements,to get the predicted results.

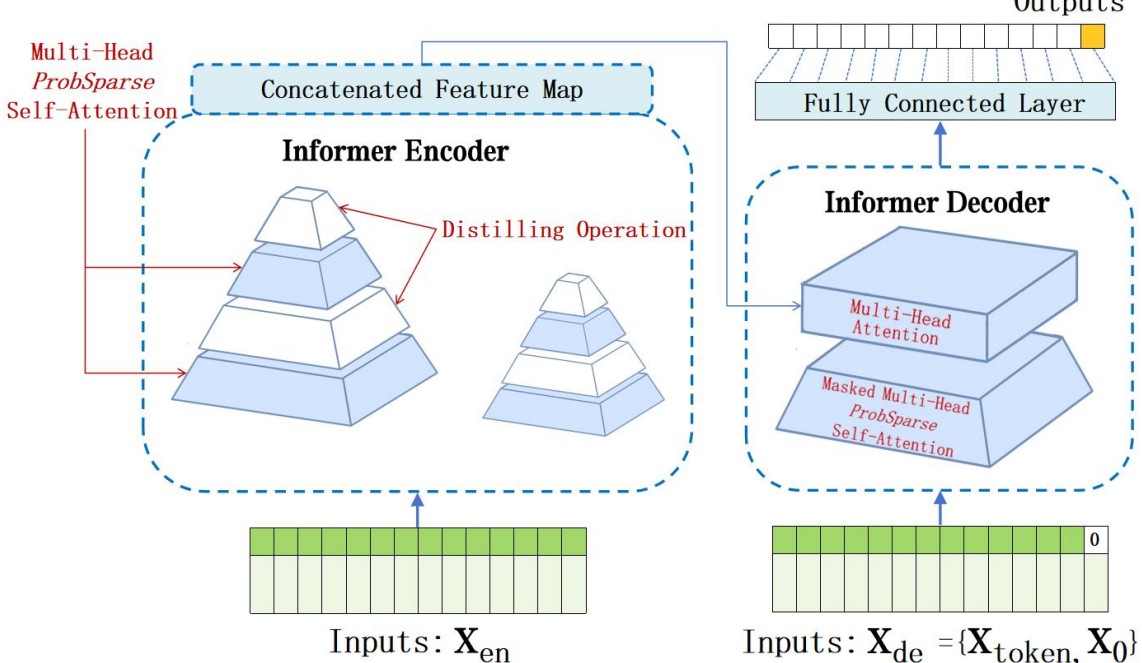

**Figure 4. Framework diagram of Informer structure including encoder (Encoder) and decoder (Decoder).**

The core idea of the attention mechanism is to compute the attention weights of each element in the input sequence,
indicating their relevance to the current output. The process first generates query (q), keyword (k), and value (v) vectors for





each element in the sequence, where the query vector represents the current network output state, where the query vector represents the current network output state, and the keyword and value vectors correspond to the input elements and their associated attribute vectors, respectively. Subsequently, these vectors are utilized to compute the attention weights via a scaled dot product relation to measure the similarity between the vectors.

Informer uses formula (10) instead of formula (9) in traditional Transformer for the calculation of attention. This enables the *ProbSparse* self-attention mechanism to significantly reduce the cost of self-attention computation while maintaining its effectiveness.

$$Attention(\boldsymbol{Q},\boldsymbol{K},\boldsymbol{V}) = softmax\left(\frac{\boldsymbol{Q}\boldsymbol{K}^T}{\sqrt{d}}\right)\boldsymbol{V} \tag{9}$$

$$Attention(\boldsymbol{Q},\boldsymbol{K},\boldsymbol{V}) = softmax\left(\frac{\bar{\boldsymbol{Q}}\boldsymbol{K}^T}{\sqrt{d}}\right)\boldsymbol{V} \tag{10}$$

$\bar{\boldsymbol{Q}}$ , $\boldsymbol{K}$ , $\boldsymbol{V}$ in Eq. (10) are the input matrices of the attention mechanism, where $d$ denotes the dimensionality of the query and key vectors (dimensionality of the inputs) and $\bar{\boldsymbol{Q}}$ in Eq. (10) contains only Top-u queries under the sparsity metric $M(\boldsymbol{q}_i,\boldsymbol{K})$ . where for $M(\boldsymbol{q}_i,\boldsymbol{K})$ an empirical method is proposed to effectively measure query sparsity, which is computed by a method similar to the Kullback-Leibler scattering. The calculation formula is as follows:

$$\overline{M}(\boldsymbol{q}_i,\boldsymbol{K}) = max\left\{\frac{\boldsymbol{q}_i\boldsymbol{k}_j^T}{\sqrt{d}}\right\} - \frac{1}{L_K}\sum_{j=1}^{L_K}\frac{\boldsymbol{q}_i\boldsymbol{k}_j^T}{\sqrt{d}} \tag{11}$$

Here for every (query) $\boldsymbol{q}_i \in \mathbb{R}^d$ and $\boldsymbol{k}_i \in \mathbb{R}^d$ in the set of key $\boldsymbol{K}$ , we have the bound as $\ln L_K \leq M(\boldsymbol{q}_i,\boldsymbol{K}) \leq max_j\left\{\boldsymbol{q}_i\boldsymbol{k}_j^T/\sqrt{d}\right\}$ $-\frac{1}{L_K}\sum_{j=1}^{L_K}\left\{\boldsymbol{q}_i\boldsymbol{k}_j^T/\sqrt{d}\right\} + \ln L_K$ .The formula also holds when $\boldsymbol{q}_i \in \boldsymbol{K}$ .

In order to allow the encoder to process longer sequential inputs under memory constraints, the self-attention Distilling operation is used in the model, which not only captures long distance dependencies in long input sequences, but also reduces the dimensionality of the model and the network parameters.



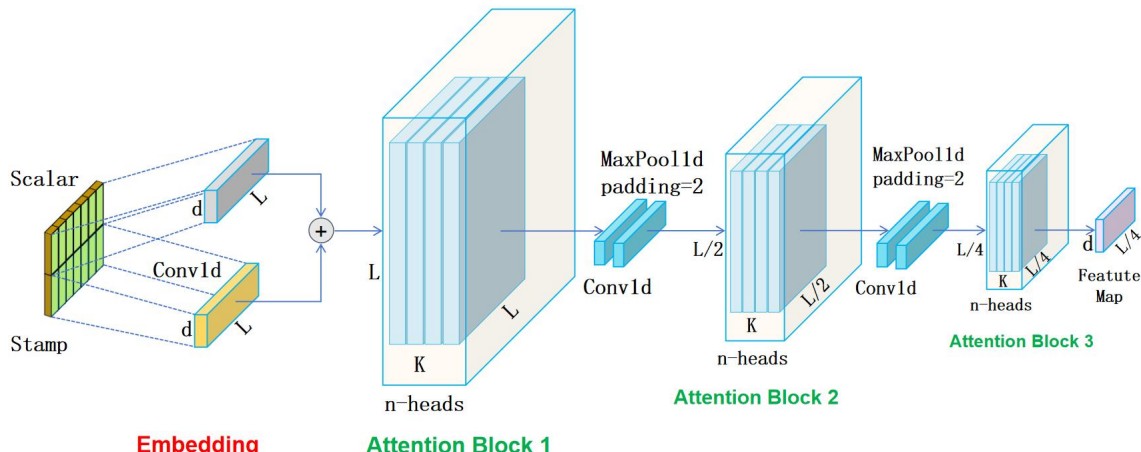

**Figure 5. Informer's encoder structure.**

As shown in Fig. 5, the encoder contains three parts such as attention block(Attention Block), convolutional layer (Conv1d) and maximum pooling layer (MaxPool) for encoding the input data. After the main stack (Attention Block 1), Attention Block 2 and Attention Block 3 halve the input successively, thus increasing the reliability of the distillation operation, and the whole process continues by gradually reducing to 1/4 of the original length. At the end of the encoder, all feature maps are concatenated and the output of the encoder is directed to the decoder. In addition, a self-attention mechanism is employed to evaluate the main features and generate a unified self-attention feature map for subsequent levels. Essentially, this process reduces the complex self-attention mechanism in the Transformer model to a simpler, smaller form that is suitable for integration into the Informer model.

In Figure 5, the decoder consists of two identical multi-head attention layers. The main difference is that predictions are generated through a process called generative inference, which greatly speeds up the long-term prediction process. The decoder provides the following vectors:

$$X_{de}^t = concat\left(X_{token}^t, X_0^t\right) \in \mathbb{R}^{\left(L_{token} + L_y\right) \times d_{model}} \tag{12}$$

where $X_{token}^t \in \mathbb{R}^{L_{token} \times d_{model}}$ is the start marker and $X_{token}^t \in \mathbb{R}^{L_y \times d_{model}}$ is a placeholder for the target sequence (set to 0). The computation of *ProbSparse* self-attention is then tuned by setting the inner product to -∞ via the masked multi-head attention mechanism, thus avoiding the involvement of each position in subsequent positions. Finally, a fully connected layer produces the final prediction.

For Informer model, mean square error (MSE) is used as a loss function for back propagation during training. A 0.2dropout is applied during training to enhance the model generation. The Adam optimizer is used to update the weights with a learning rate of 0.000001 and the number of layers of the model is 128.The number of training rounds for this model is set to 6. When the model does not decline in 3 rounds of loss, the training of the model is stopped. This configuration



allows the model to decay and effectively prevents overfitting by reducing the parameter magnitude, resulting in a better trained model.

Statistically based methods are widely used in constructing prediction intervals, and they are capable of detecting anomalies and analyzing anomalies in seismic precursors. (Zhang et al., 2023b) constructed confidence intervals for the predicted values by calculating the residual values of raw infrared long-wave radiation (OLR) data. When the data exceeded the upper and lower limits of the confidence interval, they were determined as cold or hot anomalies, respectively. (Chi et al., 2023) used the lower and upper bound estimation (LUBE) method to directly construct prediction intervals to extract anomalies in seismic precursors. (Li et al., 2024a) utilized a nonparametric method of constructing prediction intervals using

data extremes to construct upper and lower bounds of prediction intervals directly.

The upper and lower bounds of the predicted sequence output from the decoder are determined in this paper using the normal distribution method. The prediction interval of the network consists of upper bound and lower bound.. The upper and lower bounds of the prediction intervals in the network are determined using the following equation (13):

$$
\begin{aligned}
Lower &= Prediciton - Z \times mad, \\
Upper &= Prediciton + Z \times mad,
\end{aligned}
\tag{13}
$$

where Prediction is the predicted value; Z is the Z-score of the normal distribution, which is approximately 1.44 at the 85% confidence level; and mad denotes the median absolute deviation.

## 4 Data processing

The borehole strain data used in this study were provided by the Beijing Seismological Bureau. The borehole strain data of Menyuan station from January 1, 2020 to May 31, 2021 were selected for extracting the anomalous signals before the Mado

earthquake, taking into account the time of the earthquake that occurred in the Mado earthquake. The borehole strain data can effectively extract anomalies in the short and medium term. In this paper, the training and validation sets are divided into the first 8 months of data in 2020, and the prediction set is selected from the 7 months before the earthquake. The data from Mengyuan station were validated by the self-consistent equation, and the four-component borehole strain data from Mengyuan station were converted into two shear strain components $S_{13}$ and $S_{24}$ and one surface strain component $S_a$ by

using the strain conversion equation as shown in Fig. 6.

Subsequently, the $S_a$ data from the Mengyuan station were decomposed using the Segmented Variational Modal Decomposition (SVMD). Five modes are selected in the decomposition process, the decomposition parameters are set to 2000 bandwidths, and the convergence accuracy is set to 1e - 7. The decomposition results are shown in Fig. 6, where k1 represents the annual trend component, k2 represents the tidal component, and the remaining three components are summed

up to obtain the final SVMD results. The decomposition results are compared with the relevant influencing factors, and the decomposition reveals that the method effectively eliminates the influence of seasonal trends and solid tides on the borehole strain data, and significantly improves the extraction of anomalous signals.





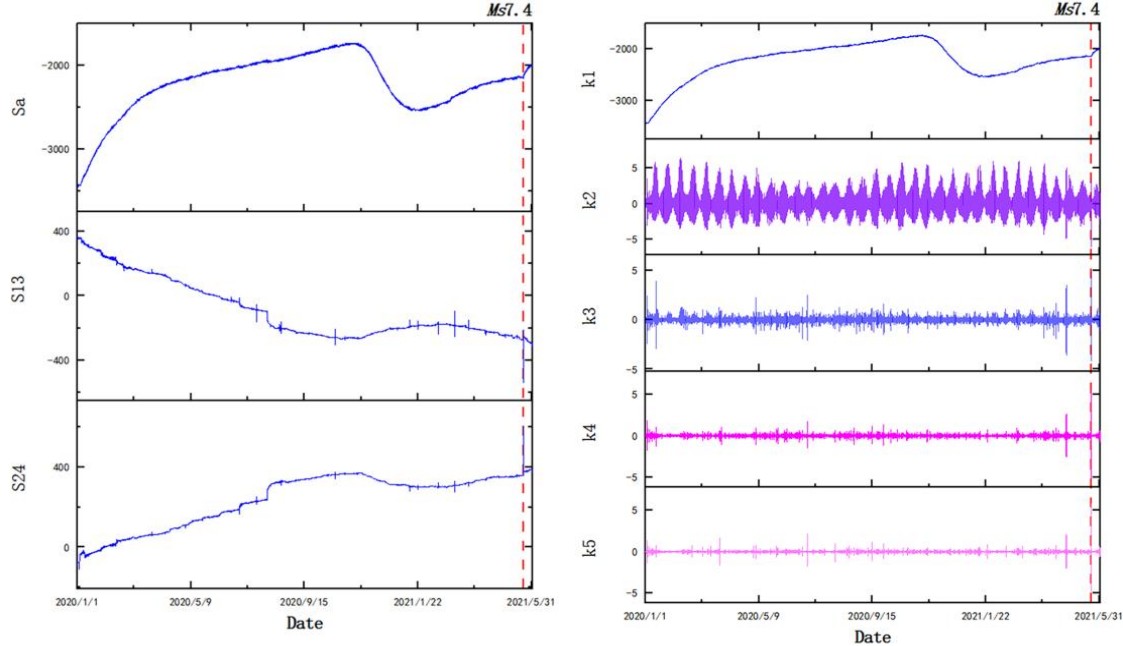

**Figure 6. Data for** $S_a$, $S_{13}$, $S_{24}$ **at the Mengyuan station and the results of the SVMD decomposition of** $S_a$. **k1 denotes the trend**
**term, k2 denotes the earth tides, and k3, k4, and k5 denote the strains associated with the earthquakes. The red dashed line**
**indicates the date of the earthquake.**

## 5 Results and Discussion

In this study, we use the SVMD-Informer network to extract the pre-seismic anomaly signals of the Mado earthquake from
the borehole strain data of the Mengyuan station. The analysis focuses on identifying pre-seismic anomalies based on the
obtained results. The anomalies were recognized when the raw data exceeded the corresponding upper or lower bounds.The
prediction results of the SVMD-Informer network are shown in Fig. 7. From the figure, it can be seen that the real values are
highly compatible with the prediction intervals, especially in the peak and valley regions of the real data, and the prediction
interval can better capture the change trend of the data. This result indicates that we have a high reliability in predicting the
borehole strain data using the SVMD-Informer network.



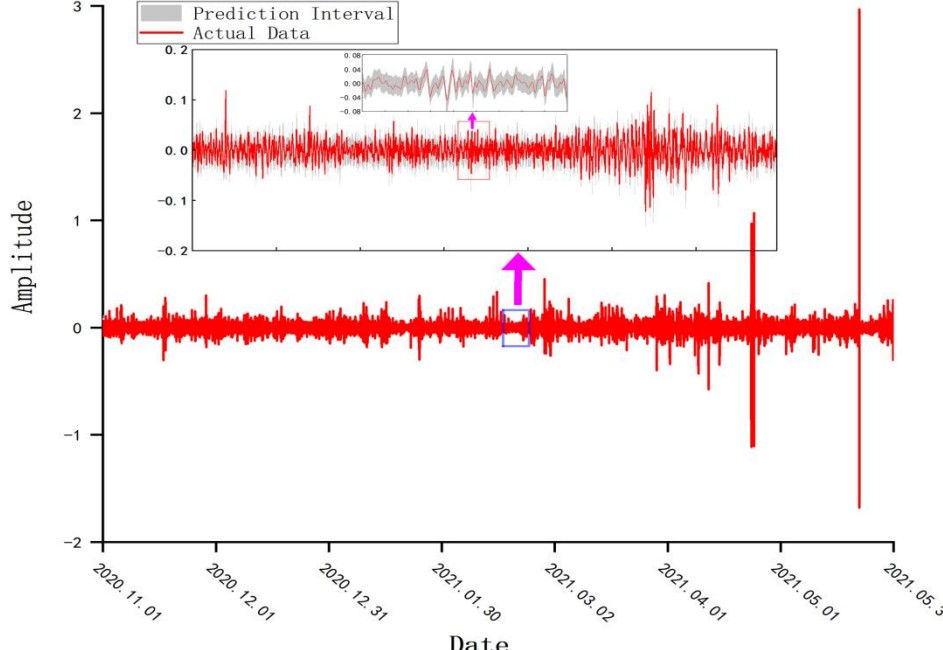


**Figure 7. Detail of the prediction results of the Informer model for the borehole strain data at the Menyuan station. The red line indicates the actual data and the gray area indicates the prediction interval.**

To identify anomalies in the prediction of borehole strain data, we used the following criteria: (a) detecting 15 points outside the interval within a 30-min window; (b) identifying interval bandwidths where the difference between the center of
the predicted interval and the actual value is more than 1.5 times the interval bandwidth (more than three such points occurring in the same 30-min period). Days that fulfill these conditions are considered anomalous (Chi et al., 2023). This criterion can effectively distinguish random fluctuations from significant anomalies, thus significantly improving the accuracy of anomaly identification and providing a more precise basis for earthquake precursor analysis.

(De Santis et al., 2017) first proposed an anomaly accumulation analysis method based on the S-shaped fitting function
by studying the Swarm satellite data of the April 25, 2015 Nepal earthquake, and found that the S-shaped fitting function performs more superiorly in describing the anomaly accumulation process compared to the traditional linear fitting method. Similarly, (Yu et al., 2024b) analyzed the geomagnetic data by using a geomagnetic model converter (ATGM) based on the self-attention mechanism and successfully extracted magnetic anomalies (MG-anomalies) in the Wenchuan, Lushan, and Kangding earthquakes, and found that the number of anomalies accumulating conformed to the S-type growth law. (Li et al.,
2024b) used Graph WaveNet to extract pre-earthquake anomalies from borehole strain data of several stations near the earthquake source, from which they found that the anomalies accumulated at several stations before the Lushan earthquake showed an S-type accelerated upward trend, and further explained their correlation with the occurrence of the Lushan earthquake. (Fan et al., 2024) used the Sigmoid function to fit the anomaly accumulation results of electron concentration, geomagnetic data, and their fusion parameters in the Mado earthquake, and the results showed that the S-shaped curves can





well reflect the process of pre-seismic acceleration and post-acceleration deceleration recovery of the earthquake. These
       results provide important theoretical support and reference basis for the anomaly accumulation analysis in this paper.

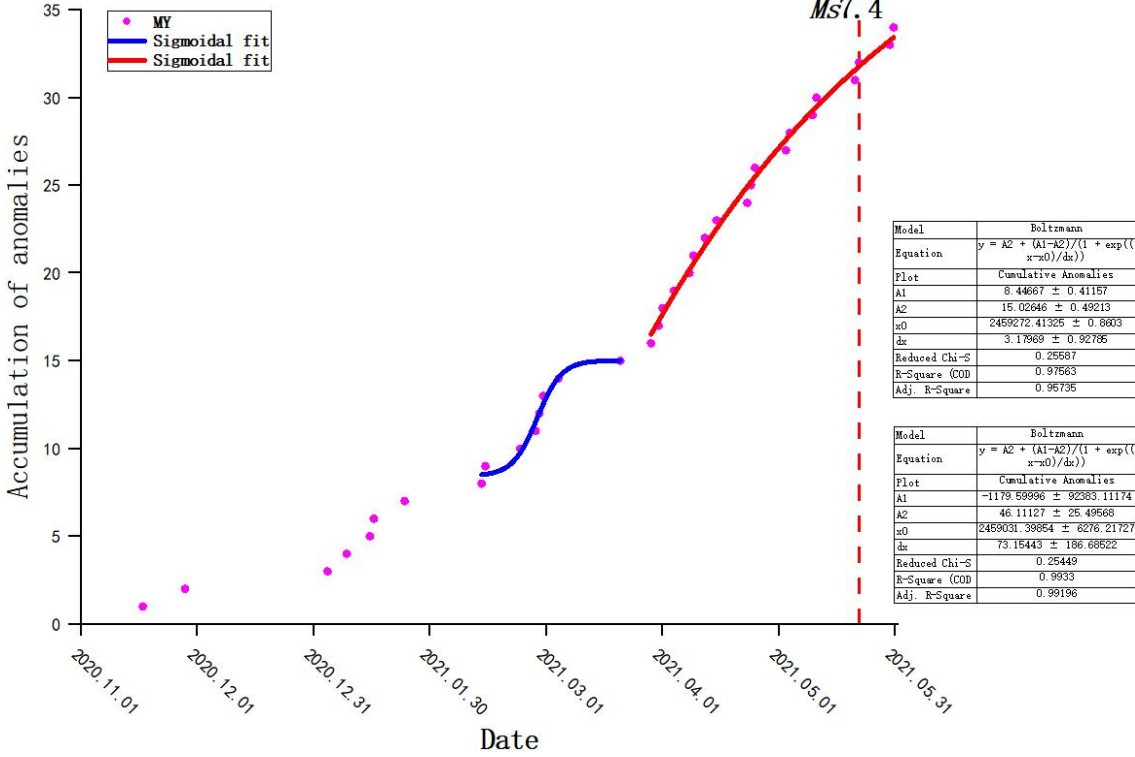

**Figure 8. Cumulative results of anomalous days of borehole data at Mengyuan station (MY). The red dashed line indicates the date
of the earthquake, and the blue and red curves indicate the results of the S-fit function for the first and second phases, respectively.**

As shown in Figure 8, the cumulative results of anomalous days at the Mengyuan station from before to the time of the

       Mado earthquake show a two-phase continuum of change. The first phase shows that the number of anomalous days from

       February 13, 2021 to the middle of the two months prior to the earthquake accelerated from the beginning of the anomaly on

       February 13 to the middle of March, and then leveled off. The second phase shows that the number of anomalous days has

       been in an accelerated increasing trend from the end of March until the earthquake. The timing of the first anomaly at

Mengyuan station coincides with the timing of the first-order anomaly of the index of microwave radiation anomaly (IMRA)

       found by (Jing et al., 2022), which both appeared in mid-February. The timing of the second anomaly coincides well with the

       outward long-wave radiation (OLR) anomalies reported by (Zhang et al., 2023a) and (Zhang et al., 2023b), and the

       observation of the thermal anomalies reveals that the Mengyuan station is located within the range of the thermal anomalies

       of March 21 and March 22, which suggests that there was interplate thermal motion on the surface at this time. Moreover,

the correlation coefficients of the geoelectric field began to show a significant decrease two months before the Mengyuan

       earthquake (Xin and Zhang, 2021). Meanwhile (Fan et al., 2024) calculated the Benionff strain S 90 days before the

       earthquake, and the results showed that the time of acceleration in the first stage was basically the same as the time when our





second stage began to accelerate, inferring that there might be frequent lithospheric activities at this time. Therefore, the anomalies we found in the borehole strain data have some reliability and research value.

In addition, the two-stage accelerated growth of the borehole strain accumulation results may reveal two critical states prior to the main earthquake. This is consistent with (Ma et al., 2014), who found that there was a sub-stabilized phase of fault deformation prior to the earthquake, which manifested itself as two instability events. It is particularly noteworthy that the acceleration of the second stage is more pronounced than the first stage, which supports their inference that the stress release is weaker in the first stage and accelerates and expands in the second stage until rupture occurs.

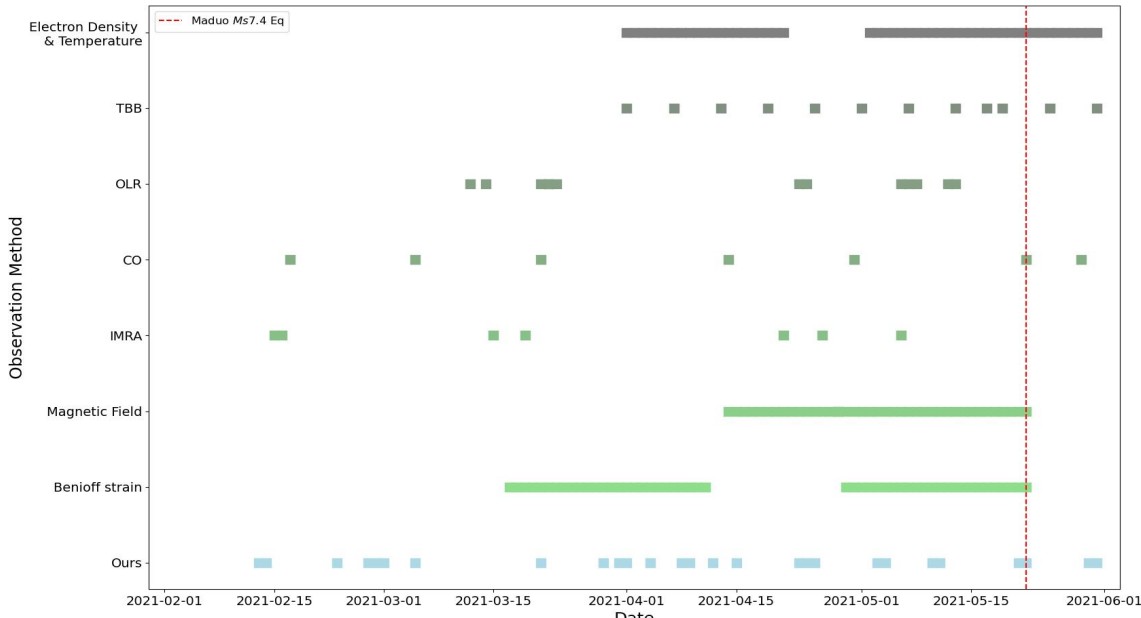


**Figure 9. Summary of our study of borehole strain anomalies (light blue) versus subsurface-to-atmosphere anomalies (from light green to gray) for the Mado Ms7.4 earthquake. The red dashed line indicates the date of the earthquake.**

As shown in figure 9, related researchers have studied a variety of anomalies that occurred prior to the Mado earthquake. From the observations at the surface, the magnetic field showed two accelerated phases, with the first phase showing an
accelerated growth trend from 38 to 24 days before the main earthquake, and the second phase showing a significant increase in the accumulation of magnetic field anomalies from the beginning of the 24th day before the main earthquake to the time of the earthquake (Fan et al., 2024). In the transition from the surface to the atmosphere, the CO concentration at the 600-hPa altitude fluctuated from February 17, increased again after a small decrease on March 5, and showed the first anomaly peak on March 21st. The maximum anomaly of near-surface CO appeared in April, with a sudden increase on April 14, and the
anomaly increased to the maximum level before and after the earthquake in late April (Shi et al., 2024). It is noteworthy that the timing of the CO anomalies is highly consistent with the timing of our borehole strain anomalies. The temperature of brightness blackbody (TBB) started to increase significantly in the direction of the fracture zone north of the epicenter about 1.5 months before the earthquake, reached its maximum intensity on May 17, and the anomaly gradually weakened during



the earthquake (Yang et al., 2024). In the atmosphere, the electron density and electron temperature showed anomalous
activity about 40 and 20 days before the earthquake (Tian et al., 2023; Fan et al., 2024; Du and Zhang, 2022). The synthesis
in Fig. 9 contains anomalies covering a wide range of phenomena from our borehole strains in the subsurface to electron
concentrations and temperatures in the atmosphere. It can be found that the borehole strain anomalies show a more
comprehensive temporal coverage in the short-term anomalies and proseismic anomalies. In this regard, our study concludes
that anomalies existed three to two months before the Mado earthquake. Therefore, we hypothesize that the borehole strain
anomalies may be related to the Mado earthquake.

## 6 Conclusion

In this study, we propose a SVMD-Informer network-based anomaly detection method for borehole strain data, aiming at
extracting pre-seismic anomaly signals, and validate it with the 2021 Mado Ms7.4 earthquake as an example. By combining
segmented variational modal decomposition (SVMD) and Informer network, we not only effectively solved the problems of
slow computation speed and memory limitation of the traditional VMD method when dealing with large-scale data, but also
significantly improved the accuracy and efficiency of the prediction of long-series time series. The SVMD method maintains
the correlation between data points through the sliding window mechanism, while the Informer network significantly reduces
the computational complexity and is able to predict long-time sequences at once through its unique *ProbSparse* self-attention
mechanism and self-attention Distilling operation, thus improving the reliability of anomaly detection.

By analyzing the borehole strain data from the Mengyuan station, we identify two phases of accelerated anomaly
accumulation before the Mado earthquake, which occurred about 3 months and 2 months before the earthquake, respectively,
and the anomaly accumulation curves show a typical S-shape growth trend. In addition, our results are highly compatible
with the time windows of other seismic precursor anomalies, such as the index of microwave radiation anomalies (IMRA),
outward long-wave radiation (OLR), and geoelectric field, which further validate the correlation between the borehole strain
anomalies and the Mado earthquake. With the continuous progress of machine learning technology and the continuous
accumulation of seismic observation data, combined with the SVMD-Informer method proposed in this paper, we are
expected to realize more accurate earthquake prediction in the future, which can provide strong support to mitigate the risk
of seismic disasters.

*Data availability.* The data set is provided by China Earthquake Networks Center, National Earthquake Data Center.
(https://data.earthquake.cn).

*Author Contributions. Conceptualization, Shanzhi Dong, Chengquan Chi and Zhichao Zhang; Data curation, Shanzhi Dong,
Chengquan Chi and Changfeng Qin; Formal analysis, Shanzhi Dong, Chengquan Chi, and Jie Zhang; Investigation,
Shanzhi Dong, Chengquan Chi and Yu Duan; Methodology, Chengquan Chi; Resources, Chengquan Chi; Software, Shanzhi*



*Dong and Chenyang Li; Supervision, Chengquan Chi; Validation, Shanzhi Dong, Chengquan Chi; Writing – original draft, Chengquan Chi and Shanzhi Dong; Writing – review & editing, Shanzhi Dong , Chengquan Chi and Zhichao Zhang.*

*Competing interests.* The authors declare that they have no conflict of interest.


*Acknowledgments.* The authors would like to thank Qiu Z. H., Tang L., and Yang D. H. from the China Earthquake Administration for giving essential help in accessing the website and downloading the strain data. Acknowledgement for the data support from " China Earthquake Networks Center, National Earthquake Data Center. (https://data.earthquake.cn)".

*Financial support.* This work was supported by the Hainan Provincial Natural Science Foundation of China under Grants 622RC669. This work is supported by the specific research fund of the Innovation Platform for Academicians of Hainan Province.

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
