# Peer review of "Research on the Extraction of Pre-Seismic Anomalies in Borehole Strain Data of the Maduo Earthquake Based on the SVMD-Informer Model"

_EGUsphere, 2025_

## Author Comment (AC1)

**Response to Reviewer 2:**

I am very grateful to your comments for the manuscript. Thank you for your advice. All your suggestions are very important. They have important guiding significance for our paper and our research work. We have revised the manuscript according to your comments. The response to each revision is listed as following:

**Comment 1**

In the first part, the authors could be more specific about the different works they cite.

**Response:**

Thanks for your suggestion.

In response to the issues you have pointed out, we have already made the appropriate changes and additions in Part I. Specifically include:

In the original manuscript 32-34 we added a detailed description of the findings of (Seropian et al., 2021), which explicitly states that most types of volcanoes are likely to be triggered by earthquakes, thus reinforcing the scientific basis for earthquake-induced natural hazards in the context of this study. Also new is the addition of (Koshimura and Shuto, 2015), which describes the devastating tsunami triggered by the Ms 9.0 magnitude earthquake that struck Japan's North Pacific coast on March 11, 2011, completely destroying many coastal communities, further emphasizing the severity of earthquake-induced secondary disasters. Finally, we modify the paragraph to read "They can damage infrastructure such as ground, transportation, and buildings, and may lead to secondary disasters such as volcanic eruptions (Seropian et al., 2021), tsunamis (Koshimura and Shuto, 2015), and landslides (Fan et al., 2019). Meanwhile, seismic hazards not only threaten human lives (Potter et al., 2015), but also have far-reaching impacts on socioeconomic development and quality of life (Peptan et al., 2023).".

In addition, in lines 104-107 of the original manuscript, we have made the citations of (Zhu et al., 2020; Yu et al., 2021; Li et al., 2024) more specific. The addition reads "In a recent study, (Zhu et al., 2020) used harmonic analysis to eliminate the effects of solid tidal and seasonal trends on borehole strain data. (Yu et al., 2021) used state-space modeling to remove the strain response due to seasonal trends, barometric pressure, solid tides, and water level variations, thus effectively isolating non-tectonic disturbances. (Li et al., 2024) successfully extracted the pre-seismic anomalies of the 7.0 magnitude earthquake in Jiuzhaigou by removing the effects of seasonal trends and tides on the borehole observations based on the variational modal decomposition (VMD) and combining with the Graph WaveNet model to process the multi-station data.".

**Comment 2**

The authors should clarify the use, operation, and scope of borehole strain gauges in terms of the signals they measure.

**Response:**

Thanks for your suggestion.

Borehole strain observation is an important means of studying crustal deformation and changes in the earth's stress field, and can observe crustal deformation under the action of a regional stress field (Qiu and Shi, 2004). Borehole strain gauges place sensors in

boreholes to observe the deformation of an extremely small part of the crust relative to the earth, which can be approximated as a point deformation observation. Inside the probe of the borehole strain gauge is the element that measures the change in internal diameter, and the probe is sealed with a sealed cylindrical sleeve and placed into the borehole, which is filled with a special cement that couples the probe to the pores of the surrounding medium.

[Figure]

**Fig. 1. Principle model of the plane strain tensor observed by a four-component borehole strain gauge.**

The new YRY-4 four-component borehole strain gauge developed independently by China has a digital sampling rate of once per minute. Figure 1 gives the principle model of the plane strain tensor observed by the four-component borehole strain gauge. The schematic model assumes linear elasticity and isotropy of each medium and is used to measure the horizontal strain state of rocks. Strain gauge $i$ in the cylinder directly measures the change in diameter of the corresponding azimuth angle $\theta_i$ caused by the change in strain state (Chi et al., 2009). The relationship equation between the measured value $s_i$ and the strain change $(\varepsilon_1, \varepsilon_2, \varphi)$ is shown below:

$$s_i = A(\varepsilon_1 + \varepsilon_2) + B(\varepsilon_1 - \varepsilon_2)cos2(\theta_i - \varphi) \tag{1}$$

where $\varepsilon_1$ and $\varepsilon_2$ are the maximum and minimum principal strains, respectively, $\varphi$ is the principal direction, and $A$ and $B$ are the two parameters to be determined.

The relative change in aperture measurements of an arbitrarily selected element, denoted as $S_1$, is rotated clockwise by 45° in turn, and there are element measurements $S_2$, $S_3$, and $S_4$ (Qiu et al., 2021). According to equation (1), the four-element observations are:

$$\begin{cases} s_1 = s_{\theta_1} = A(\varepsilon_1 + \varepsilon_2) + B(\varepsilon_1 - \varepsilon_2)cos2(\theta_1 - \varphi) \\ s_2 = s_{\theta_{1+\pi/4}} = A(\varepsilon_1 + \varepsilon_2) - B(\varepsilon_1 - \varepsilon_2)sin2(\theta_1 - \varphi) \\ s_3 = s_{\theta_{1+\pi/2}} = A(\varepsilon_1 + \varepsilon_2) - B(\varepsilon_1 - \varepsilon_2)cos2(\theta_1 - \varphi) \\ s_4 = s_{\theta_{1+3\pi/4}} = A(\varepsilon_1 + \varepsilon_2) + B(\varepsilon_1 - \varepsilon_2)sin2(\theta_1 - \varphi) \end{cases} \tag{2}$$

$s_i (i = 1,2,3,4)$ are the measured values from the four instruments and the YRY-4 four-component borehole strain gauge has good data self-checking (Su, 2019). It should be available when the coupling between the probe and the surrounding rock is in ideal condition:

$$s_1 + s_3 = s_2 + s_4 \tag{3}$$

Equation (3) is the self-consistent equation for the YRY-4 four-component borehole strain gauge, which is considered reliable when the data satisfy the above results.

Borehole strain gauges observe the amount of change in strain, so the terms in the self-consistent equation are also the amount of change. In practice, due to the coupling of the borehole strain gauge with the surrounding rock layer and the instrument itself will make each component of the observed data to produce a certain drift phenomenon, that is, the annual trend in the data, $S_1 + S_3$ and $S_2 + S_4$ in the numerical value is not equal, but the two curves of the form is the same, so the self-consistency equation can be written in the form of the formula (4).

$$S_1 + S_3 = k(S_2 + S_4) \tag{4}$$

where $k$ is the self-consistency coefficient and the data are considered to satisfy self-consistency when $k \geq 0.95$.

Due to the nature of strain observation in boreholes, it is necessary to couple the probe to the surrounding medium in order to carry out strain observations, and the medium in which the probe is installed must be continuous and uniform in order to meet the quality requirements. Because of the surface will exist from the air pressure changes and human activities and other interference, so the probe needs to be installed in a certain depth of the ground. Our four-component borehole strain gauges are installed at a depth of about 40 meters below the ground surface, and the data quality of our four-component borehole strain observations is satisfactory (Qiu et al., 2021).

**Comment 3**

The paragraph "By mounting strain gauges deeper in the bedrock, borehole strain gauges are able to continuously record stress and strain data, making them a key tool for monitoring crustal deformation" is unclear since it does not specify what type of borehole strain gauge they use and how they ensure the results mentioned. Furthermore, it does not specify how they physically justify the scaling suggested in the paragraph: "The high-resolution recordings provided by borehole strain gauges allow us to capture small changes in strain, thus providing accurate data to support a deeper understanding of crustal deformation processes."

**Response:**

Thanks for your suggestion.

(1) The paragraph "By mounting strain gauges deeper in the bedrock, borehole strain gauges are able to continuously record stress and strain data, making them a key tool for monitoring crustal deformation" is unclear since it does not specify what type of borehole strain gauge they use and how they ensure the results mentioned.

In the study of (Qiu et al., 2013), it was noted that the YRY-4 four-component borehole strain gauge was installed at a depth of about 40 m, and the sampling rate was 1 sample per minute. The unique design of the YRY-4 four-component borehole strain instrument allows quantitative estimation of the confidence of the data by means of a self-consistency test, without resorting to earth tides or any special signals. In addition, the study proposed a relative correction method for norm sensitivity and demonstrated its effectiveness in improving data confidence. Therefore, this confirms that the YRY-4 four-component borehole strain gauge has the ability to continuously monitor the stressstrain changes in the earth's crust, which provides reliable data support for earthquake prediction and tectonic movement studies. Also in response to your comments, we note that the statement in lines 64-65 of the original manuscript does contain a lack of specificity in that it does not specify the type of borehole strain gauges used and does not guarantee the results described above. Therefore, we modify and supplement this paragraph. It should be revised as "As the main observation equipment of China's digital seismic observation network, China's self-developed YRY-4 four-component borehole strain gauge is usually installed at the bottom of bedrock at 40 meters, and has the capability of minute-level strain sampling, which can continuously record high-resolution stress and strain changes (Qiu et al., 2013).".

(2) Furthermore, it does not specify how they physically justify the scaling suggested in the paragraph: "The high-resolution recordings provided by borehole strain gauges allow us to capture small changes in strain, thus providing accurate data to support a deeper understanding of crustal deformation processes.".

Borehole strain gauges have the advantages of high sensitivity, broad bandwidth, and long-term stability (Lou and Tian, 2022). Its high-resolution recordings are able to clearly observe strain solid tides, seismic strain waves, and high-frequency microseismicity from microfractures in the formation. This observational capability stems from the sensor's high sensitivity and low-noise design, allowing it to cover a wide band of signals ranging from long-term slow deformation to high-frequency seismic waves. Based on these characteristics, the borehole strain gauge can capture small strain changes and thus provide accurate data support for in-depth studies of crustal deformation processes. Therefore, we revise lines 65-67 of the original manuscript to read "The high-resolution recordings provided by borehole strain gauges allow us to capture minute strain changes, thus providing accurate data to gain insight into crustal deformation processes (Lou and Tian, 2022).".

In addition, to ensure sentence coherence and comprehensiveness, we revised the original manuscript 62-72 to read "Borehole strain observation is superior to GPS and laser strain gauges in capturing short- and medium-term strain changes and pre-seismic anomalies (Qiu and Shi, 2004), and is an important means to study crustal deformation and stress field changes. As the main observation equipment of China's digital seismic observation network, China's self-developed YRY-4 four-component borehole strain gauge is usually installed at the bottom of bedrock at 40 meters, and has the capability of minute-level strain sampling, which can continuously record high-resolution stress and strain changes (Qiu et al., 2013). The high-resolution recordings provided by borehole strain gauges allow us to capture minute strain changes, thus providing accurate data to gain insight into crustal deformation processes (Lou and Tian, 2022). In addition, the borehole strain gauge not only provides four-component data, but also records ancillary observations such as solid tides, air temperature, and air pressure (Chi, 2009; Tang et al., 2023).".

**Comment 4**

On line 80, they should specify the methodology that led to the discovery of precursors by analyzing eigenvectors and eigenvalues. Where did they get it?

**Response:**

Thanks for your suggestion.

(Zhu et al., 2020) conducted an anomaly extraction study on the pre-earthquake borehole strain data of Wenchuan earthquake by using principal component analysis (PCA) method to analyze the first eigenvalue and the first eigenvector of the borehole strain data, and to extract the anomalous features of the strain changes before the Wenchuan earthquake. The eigenvalue indicates the main intensity of the signal, and the anomalies may imply stress changes or earthquake precursors. The eigenvectors indicate the directional characteristics of the strain changes, and the changes in their spatial distribution can reveal the evolution of the fault from a steady state to a sub-instable state. Their results show that the borehole strain gauges recorded the preparation stage of the Wenchuan earthquake, and the principal component analysis can effectively extract the crustal strain change characteristics. Therefore, we revised lines 79-80 of the original manuscript to read "(Zhu et al., 2020) studied the anomalous characteristics of the borehole strain data before the Wenchuan earthquake by using principal component analysis. By analyzing the first eigenvalue and the first eigenvector of the borehole strain data, the characteristics of pre-earthquake crustal strain changes are revealed.".

**Comment 5**

Correct Maduo instead of Mado on lines 134, 140, 180, 335, and elsewhere.

**Response:**

Thanks for your suggestion.

We have corrected the original manuscript text from the incorrect Mado to the correct Maduo.

**Comment 6**

The authors used the SVMD-Informer network to extract preseismic anomaly signals from the Mado earthquake from well deformation data at Mengyuan Station. The authors state that anomalies associated with the earthquake were recognized when the raw data exceeded the corresponding upper or lower limits. The question is: What is the physical basis for determining the criteria described in paragraphs 366 to 368?

**Response:**

Thanks for your suggestion.

The anomaly criteria presented in paragraphs 366-368 of the original manuscript build on prediction intervals constructed using the SVMD-Informer model. Although the form of these criteria is statistical in nature, we believe that they reflect to some extent the processes associated with earthquake gestation and have some physical significance.

1. Physical interpretation of the prediction intervals: The 85% confidence intervals that we have constructed represent the range of normal fluctuations in crustal strains of the tectonic system under steady-state conditions. This range essentially describes the strain behavior "under normal tectonic evolution". Observations outside this range indicate that the crustal system may have deviated from elastic equilibrium, which may be caused by processes such as microfracture extension, localized stress concentrations, or

pre-seismic nonlinear evolution.

2. Physical significance of the anomalous day criterion: We define two criteria for anomalies: (a) ≥15 anomalies outside the prediction interval in a 30-minute time window; and (b) ≥3 points where the actual value deviates from the center of the prediction interval by more than one and a half times the width of the interval in the same time window. These criteria are used to identify strain deviation behaviors with temporal aggregation, persistence, and nonlinear characteristics that are closely related to pre-seismic system destabilization and rapid energy aggregation. When the crustal system is close to rupture, it tends to exhibit nonlinear dynamics such as increased fluctuations and enhanced system response.

3. The pre-seismic anomaly accumulation process presented in Fig. 8 of the original manuscript shows a typical "S-type two-phase acceleration" pattern, which further suggests that these anomalies are not noise, but may reflect key evolutionary stages in the proseismic process. This phenomenon has been recognized in several studies on seismic phase transition theory (e.g., (De Santis et al., 2017; Fan et al., 2024)).

Therefore, although these anomaly detection criteria are set on the basis of statistical modeling, we believe that they reflect, to some extent, the processes associated with earthquake gestation and have some physical significance.

[Figure]

**Figure 8 of the original manuscript. Cumulative results of anomalous days of borehole data at Mengyuan station (MY). The red dashed line indicates the date of the earthquake, and the blue and red curves indicate the results of the S-fit function for the first and second phases, respectively.**

**Comment 7**

If an earthquake is considered to correspond to a phase transition, are the results shown in this analysis associated with the critical state or preparation mechanism of the seismic process?

**Response:**

Thanks for your suggestion.

The two-stage accelerated growth of the borehole strain accumulation results may reveal two preparatory mechanisms prior to the mainshock. This is consistent with the theory of fault synergistic process of (Ma et al., 2014). They found that the occurrence of earthquakes is closely related to the three-stage synergistic evolution of faults through an indoor experimental study of plane-walk-slip faults. In the first stage, the initial stress nonlinear divergence leads to localized weakening and the formation of discrete strain release zones. The second stage is characterized by an increase in stress and a widening of the strain release zone. In the final stage, the expansion of the strain-release region and the rapid increase of the strain level in the strain-accumulation region. The anomalous cumulative acceleration about 3 months before the Maduo earthquake corresponds to the first and second stages in the theory, which is manifested by the deviation of the stress curve from linearity and the beginning of the formation and slow expansion of the discrete release zone. The acceleration 2 months before the earthquake reflects the characteristic changes of the third stage, which is characterized by the accelerated expansion of the release zone and the sharp increase of the strain in the accumulation zone. Therefore, we believe that the anomalies observed before the Maduo earthquake are related to the process of earthquake incubation.

**Comment 8**

The conclusion section must be improved

**Response:**

Thanks for your suggestion.

The conclusion of the original manuscript lacks a sense of hierarchy, has redundant sentences and lacks synthesis. Therefore, we modify the conclusion to read "In this study, a SVMD-Informer network-based anomaly detection method for borehole strain data is proposed, and the 2021 Maduo Ms7.4 earthquake is used as an example for pre-seismic anomaly extraction. The method optimizes the problems of slow computation speed and memory limitation existing in the traditional VMD by SVMD, and significantly improves the accuracy and stability of long series time series prediction by combining with Informer network. By analyzing the borehole strain data from Mengyuan station, we successfully extracted the anomalous cumulative acceleration phenomenon in the two stages before the Maduo earthquake, which appeared about 3 months and 2 months before the earthquake, respectively, and the anomalous cumulative curves showed a typical S-shape growth trend, and this result is consistent with the theory of fault synergistic process of (Ma et al., 2014). In addition, our results are highly consistent with the time windows of other seismic precursor anomalies, such as the index of microwave radiation anomalies (IMRA), outward long-wave radiation (OLR), and geoelectric field, which further validates the correlation between the borehole strain anomalies and the Maduo earthquake. With the continuous progress of machine learning technology and the continuous accumulation of seismic observation data, this method is expected to provide a higher precision technical support for earthquake prediction and help reduce the risk of seismic disasters.". The above

modifications effectively reduce repetitive descriptions, making the conclusions more coherent and concise.

**References**

Chi, S.: China's component borehole strainmeter network, Earthquake Science, 22, 579-587, 10.1007/s11589-009-0579-z, 2009.

Chi, S., Chi, Y., Deng, T., Liao, C., Tang, X., and chi, L.: The Necessity of Buiding National Strain-Observation Network from the Strain Abnormality Before Wenchuan Earthquake, Recent Developments in Word Seismology, 2009.

De Santis, A., Balasis, G., Pavón-Carrasco, F. J., Cianchini, G., and Mandea, M.: Potential earthquake precursory pattern from space: The 2015 Nepal event as seen by magnetic Swarm satellites, Earth and Planetary Science Letters, 461, 119-126, 10.1016/j.epsl.2016.12.037, 2017.

Fan, M., Zhu, K., De Santis, A., Marchetti, D., Cianchini, G., Wang, T., Zhang, Y., Zhang, D., and Cheng, Y.: Exploration of the 2021 Mw 7.3 Maduo Earthquake by Fusing the Electron Density and Magnetic Field Data of Swarm Satellites, IEEE Transactions on Geoscience and Remote Sensing, 62, 1-24, 10.1109/tgrs.2024.3361875, 2024.

Fan, X., Scaringi, G., Korup, O., West, A. J., van Westen, C. J., Tanyas, H., Hovius, N., Hales, T. C., Jibson, R. W., Allstadt, K. E., Zhang, L., Evans, S. G., Xu, C., Li, G., Pei, X., Xu, Q., and Huang, R.: Earthquake-Induced Chains of Geologic Hazards: Patterns, Mechanisms, and Impacts, Reviews of Geophysics, 57, 421-503, 10.1029/2018rg000626, 2019.

Koshimura, S. and Shuto, N.: Response to the 2011 Great East Japan Earthquake and Tsunami disaster, Philos Trans A Math Phys Eng Sci, 373, 10.1098/rsta.2014.0373, 2015.

Li, C., Qin, C., Zhang, J., Duan, Y., and Chi, C.: Analysis of Borehole Strain Anomalies Before the 2017 Jiuzhaigou Ms7.0 Earthquake Based on Graph Neural Network, EGUsphere, 10.5194/egusphere-2024-2025, 2024.

Lou, J. and Tian, J.: Review on seismic strain-wave observation based on high-resolution borehole strainmeters, Progress in Geophysics, 2022.

Ma, J., Guo, Y., and Sherman, S. I.: Accelerated synergism along a fault: A possible indicator for an impending major earthquake, Geodynamics & Tectonophysics, 5, 387-399, 10.5800/gt-2014-5-2-0134, 2014.

Peptan, C., Holt, A. G., Iana, S. A., Sfinteș, C., Iov, C. A., and Mărcău, F. C.: Considerations of the Impact of Seismic Strong Ground Motions in Northern Oltenia (Romania) on Some Indicators of Sustainable Development Characterization of the Region from a Security Perspective, Sustainability, 15, 10.3390/su151712865, 2023.

Potter, S. H., Becker, J. S., Johnston, D. M., and Rossiter, K. P.: An overview of the impacts of the 2010-2011 Canterbury earthquakes, International Journal of Disaster Risk Reduction, 14, 6-14, 10.1016/j.ijdrr.2015.01.014, 2015.

Qiu, Z. and Shi, Y.: The development status of borehole strain observation abroad, Acta Seismologica Sinica, 2004.

Qiu, Z., Kan, B., and Tang, L.: Conversion and use of four component borehole strain observation data, Earthquake, 2021.

Qiu, Z., Tang, L., Zhang, B., and Guo, Y.: In situ calibration of and algorithm for strain monitoring using four-gauge borehole strainmeters (FGBS), Journal of Geophysical Research: Solid Earth, 118, 1609-1618, 10.1002/jgrb.50112, 2013.

Seropian, G., Kennedy, B. M., Walter, T. R., Ichihara, M., and Jolly, A. D.: A review framework of how earthquakes trigger volcanic eruptions, Nature Communications, 12, 10.1038/s41467-021-21166-8, 2021.

Su, K.: Analysis of Surface Strain and Shear Strain from Four Component Borehole Strain Observation Data, Research in Shanxi, 2019.

Tang, L., Qiu, Z., Fan, J., and Yin, Z.: The apparent focal depth, emergence angle, and take-off angle of seismic wave measured by YRY-4-type borehole strainmeter as one kind of strain seismograph, Frontiers in Earth Science, 11, 10.3389/feart.2023.1036797, 2023.

Yu, Z., Zhu, K., Hattori, K., Chi, C., Fan, M., and He, X.: Borehole Strain Observations Based on a State-Space Model and ApNe Analysis Associated With the 2013 Lushan Earthquake, IEEE Access, 9, 12167-12179, 10.1109/access.2021.3051614, 2021.

Zhu, K., Chi, C., Yu, Z., Fan, M., Li, K., and Sun, H.: The characteristics analysis of strain variation associated with Wenchuan earthquake using principal component analysis, Annals of Geophysics, 63, 10.4401/ag-7946, 2020.

---

## Author Comment (AC2)

**Response to Reviewer 1:**

I am very grateful to your comments for the manuscript. Thank you for your advice. All your suggestions are very important. They have important guiding significance for our paper and our research work. We have revised the manuscript according to your comments. The response to each revision is listed as following:

**General Comment**

Since the M6.9 Menyuan earthquake occurred close to the Menyuan station, I would like to ask: why do the authors interpret the signals recorded at this borehole strain gauge station as precursors to the Maduo (Mado) earthquake, which occurred more than 400 km away? Unless the authors can clearly explain how this signal can be attributed to the Maduo earthquake rather than to unrelated phenomena or local seismic events, I find the current conclusions unconvincing.

**Response:**

Thanks for your suggestion.

Borehole strain data studies have focused on short-term as well as proseismic anomalies (3 months before the earthquake) (Chi et al., 2023; Yi, 2005; Li et al., 2024b). The anomalies of the Maduo earthquake extracted from the Mengyuan station in our study were about 3 months before the earthquake (mid-February 2021) and 2 months before the earthquake (end-March 2021), which are short-term as well as proseismic anomalies. On the other hand, the Menyuan earthquake occurred on January 08, 2022, which is about 11 months different from the time when the Maduo seismic anomaly was extracted from the Menyuan station, and it does not belong to short-term as well as proseismic anomaly. Therefore, we believe that the anomaly extracted from the Mengyuan station is a pre-seismic precursor of the Maduo earthquake, not of the Mengyuan earthquake. I will answer the details below.

**Specific Comments:**

**Comment 1**

Introduction: The section cites too many references, each describing different precursor phenomena from strain gauges, making the narrative overly verbose and lacking synthesis. Please consider summarizing the key findings of these previous studies to provide a more coherent and concise background.

**Response:**

Thanks for your suggestion.

Based on your comments, we are revising the introduction section. Previous studies are summarized to ensure more coherent and concise sentences. At the same time, we reduced the redundancy of the sentences to improve the coherence and comprehensiveness of the narrative. First, we deleted lines 37-41 of the original manuscript to remove the redundancy of sentences. At the same time, we modified lines 41-43 of the original manuscript to ensure the coherence of the sentence, and modified it to read "Researchers around the world have now explored a wide range of phenomena

before and after earthquakes, covering different structural levels of the Earth, including the subsurface, surface and atmospheric realms.". Second, we summarized the references in lines 45-50 of the original manuscript, revising them to "(Liu et al., 2023) used outward longwave radiation (OLR) data to find that thermal infrared anomalies synchronized with the tidal stress cycle preceded the 2023 M7.8 earthquake in Turkey, possibly reflecting the thermal response of tectonic stresses as they accumulate to a critical state.". In addition, revise lines 50-53 of the original manuscript to read "(Guo et al., 2015) found significant anomalous ionospheric disturbances prior to the April 11, 2012 Sumatra Ms8.6 and Mexico Ms6.7 earthquakes using global total ionospheric electron content (TEC) data.". We reduced the introduction of references in lines 53-58 of the original manuscript, as well as summarized the referenced research to ensure sentence brevity, revising it to "In addition, other scholars have also studied fields such as geomagnetism (Li et al., 2019), microwave bright temperature (MBT) (Qi et al., 2021), CO (Cui et al., 2024), and electron density (Han et al., 2023). The above studies provide abundant data support and theoretical basis for the exploration of earthquake precursors, and are of great significance to our understanding of earthquake mechanisms and potential impacts.".

Lines 83-85 of the original manuscript were deleted due to the citation of (Li et al., 2024a) in line 104 of the original manuscript, which describes the removal of seasonal trends and tidal influence on borehole observations based on Variational Modal Decomposition (VMD). Therefore, we delete lines 83-85 of the original manuscript, and we revise lines 104-107 of the original manuscript to read "(Li et al., 2024a) successfully extracted the pre-seismic anomalies of the 7.0 magnitude earthquake in Jiuzhaigou by removing the effects of seasonal trends and tides on the borehole observations based on the variational modal decomposition (VMD) and combining with the Graph WaveNet model to process the multi-station data.". Lines 111-120 of the original manuscript are too redundant, so we summarize them and modify them to read "Variational Modal Decomposition (VMD), as an adaptive signal decomposition method, is able to effectively extract the features of nonlinear and nonsmooth signals in the frequency domain, which is widely used in the analysis of complex waveforms such as steps, jumps, and burrs, and outperforms the traditional Empirical Modal Decomposition (EMD) and its derivatives in seismic signal processing (Rao et al., 2024; Li et al., 2018; Xue et al., 2019). However, with the increasing size and complexity of observed data, VMD has limitations such as high memory overhead in large-scale data processing.". Finally, lines 126-128 of the original manuscript introduced too many references; we have retained only the more recent references, while ensuring that the references introduced are recent studies.

**Comment 2**
Line 64: Could you provide more details about the borehole strain gauges used to record earthquake precursor signals, such as installation depth, sampling frequency, and observation duration?
**Response:**
    Thanks for your suggestion.

In the study of (Qiu et al., 2013), it was noted that the YRY-4 four-component borehole strain gauge was installed at a depth of about 40 m, and the sampling rate was one sample per minute. The unique design of the YRY-4 four-component borehole strain gauge allows quantitative estimation of the confidence of the data through self-consistency tests without the use of earth tides or any special signals. In addition, the study presents a relative correction method for norm sensitivity and demonstrates its effectiveness in improving data confidence. Therefore, this confirms that the YRY-4 four-component borehole strain gauge has the ability to continuously monitor the stress-strain changes in the earth's crust, which provides reliable data support for earthquake prediction and tectonic movement studies.

In addition, based on your comments we found that because lines 64-65 of the original manuscript did not specify what type of borehole strain gauges were used, we did not provide more details about the borehole strain gauges used to record the seismic precursor signals. However, we have adjusted it so that after presenting the type of borehole strain gauges used, we provide more more details about the borehole strain gauges. This makes our article, able to be more explicit. Therefore, we modify and supplement this paragraph. It should be revised as "As the main observation equipment of China's digital seismic observation network, China's self-developed YRY-4 four-component borehole strain gauge is usually installed at the bottom of bedrock at 40 meters, and has the capability of minute-level strain sampling, which can continuously record high-resolution stress and strain changes (Qiu et al., 2013).".

**Comment 3**

Line 85: Please elaborate on the cited literature. For example, what findings did Liu et al. (2014) present? What specific anomalies were observed before and after the Lushan earthquake? Did the PBO program detect reliable precursory signals for particular earthquakes?

**Response:**

Thanks for your suggestion.

(1)What findings did Liu et al. (2014) present? What specific anomalies were observed before and after the Lushan earthquake?

(Liu et al., 2014) analyzed the observation data of four-component borehole strain gauges at Guzantai before and after the 2013 Lushan Ms7.0 earthquake by the S-transform method, and found that two clusters of significant high-energy anomalous signals appeared in the time-frequency domain. The first cluster of anomalies appeared in October 2012 and lasted for about 4 months; the second cluster of anomalies appeared from a few days before to after the earthquake, in which short-period signals decayed rapidly after the earthquake. By analyzing the effects of teleseismic co-seismic effect and construction disturbances around the station, they concluded that these anomalous signals are not related to external disturbances, and may be associated with the seismic process of the Lushan earthquake. The anomalous signals gradually decayed after the earthquake, and basically disappeared by the end of August 2013, and the strain records were restored to the normal solid-tide pattern. Compared with the strain changes before and after the 2008 Wenchuan Ms8.0 earthquake, the anomalous

signals before and after the Lushan earthquake have a shorter duration and more discrete energy clusters, reflecting the differences in the strain evolution characteristics of different earthquakes. According to your suggestion, we have revised lines 87-89 of the original manuscript to read "(Liu et al., 2014) used the S-transform method to analyze the observation data of four-component borehole strain gauges at Guzantai before and after the Lushan Ms7.0 earthquake, and found that two clusters of high-energy clusters in the time-frequency domain may be related to the seismic activity in Lushan.".

(2) Did the PBO program detect reliable precursory signals for particular earthquakes? The U.S. Plate Boundary Observation (PBO) program consists of three components: a backbone of GPS receivers for general characterization of the entire plate boundary region. Observation units of GPS, borehole strain gauges, and laser strain gauges installed on tectonic zones in the western U.S. and southern Alaska. GPS units in areas not covered by installed stations (Ouyang, 2011). The seismic signals recorded by the PBO program have shown good reliability in terms of co-seismic response (Chen and Lü, 2019). Although the PBO program has achieved some results in the detection of

seismic precursor signals, there is no conclusive evidence that it has detected reliable precursor signals for specific earthquakes. Meanwhile, due to the complexity of detecting and recognizing earthquake precursor signals, the current research is still in the exploratory stage (Thomas et al., 2011).

**Comment 4**

Line 79: You mention that "Zhu et al. (2020) observed the Wenchuan earthquake precursors by analyzing the eigenvalues and eigenvectors of the borehole strain data." Could you clarify how these mathematical features are physically related to pre-seismic processes?

**Response:**

Thanks for your suggestion.

(Zhu et al., 2020) conducted an anomaly extraction study on the pre-earthquake borehole strain data of Wenchuan earthquake by using principal component analysis (PCA) method to analyze the first eigenvalue and the first eigenvector of the borehole strain data, and to extract the anomalous features of the strain changes before the Wenchuan earthquake. The eigenvalue indicates the main intensity of the signal, and the anomalies may imply stress changes or earthquake precursors. The eigenvectors indicate the directional characteristics of the strain changes, and the changes in their spatial distribution can reveal the evolution of the fault from a steady state to a sub-stable state. The results of their analysis indicate that the spatial distribution of the eigenvectors and the accelerated occurrence of eigenvalue anomalies represent the stress evolution characteristics of the faults from steady state to sub-stable state in the rock experiments. Preliminary inferences suggest that this process may also be related to the preparation stage of a large earthquake. At the same time, they show that because the principle of crustal motion is complex, it is difficult to determine the physical significance of the eigenvalues and eigenvectors when applying principal component

analysis to the study of strain earthquake precursors. The physical significance of eigenvalues and eigenvectors needs to be further explored.

Therefore, we revised lines 79-80 of the original manuscript to read "(Zhu et al., 2020) studied the anomalous characteristics of the borehole strain data before the Wenchuan earthquake by using principal component analysis. By analyzing the first eigenvalue and the first eigenvector of the borehole strain data, the characteristics of pre-earthquake crustal strain changes are revealed.".

**Comment 5**

Line 121: What does SVMD stand for? Also, the manuscript lacks a clear explanation of how SVMD improves upon conventional VMD. Please provide comparative analysis in terms of computational efficiency, memory usage, or signal decomposition performance.

**Response:**

Thanks for your suggestion.

(1) What does SVMD stand for?

SVMD in line 121 of the original manuscript denotes Segmented Variational Modal Decomposition. As shown in Fig. 3 of the original manuscript, the SVMD method is based on the principle that we choose a sliding window mechanism with a size of 7 days and a sliding step of 1 day for data segmentation. First we set the initial window to all data from day one to day seven and perform Variational Modal Decomposition (VMD) on the data within that window. Starting from the second sliding window, only the results of the VMD decomposition of the current window are retained and superimposed with the decomposition results of the previous window, and in this logical order, the data are processed sequentially to finally obtain the complete data set after SVMD processing. Therefore, we have revised line 121 of the original manuscript to add the full form for the first time in the introduction and revised it to "Segmented Variational Modal Decomposition (SVMD)".

[Figure]

**Figure 3 of the original manuscript. Schematic diagram of the Segmented Variational Modal Decomposition (SVMD).**

(2) Also, the manuscript lacks a clear explanation of how SVMD improves upon conventional VMD. Please provide comparative analysis in terms of computational efficiency, memory usage, or signal decomposition performance.

In response to your question, we provide a detailed supplement and a comparative analysis of Segmental Variational Modal Decomposition (SVMD) and Variational Modal Decomposition (VMD).

Variational Modal Decomposition (VMD) has significant advantages in dealing with nonlinear and nonsmooth signals. However, using the VMD method for global search and solving the variational problem may pose computational challenges due to the large amount of data involved, such as slow processing speed and computer memory constraints, and the use of traditional data segmentation methods may result in the loss of data correlation between neighboring data segments. In order to solve these problems, we used a Segmented Variational Modal Decomposition (SVMD) method (Chi et al., 2023).

As shown in Table 1, we choose 7-day, 15-day and 30-day data for decomposition. The experimental results show that the SVMD method is better than the traditional VMD method in terms of computational efficiency and memory usage. Running device name: DESKTOP-FPCL3PM; Running device version: Windows 11 Professional; Running device processor: 12th Gen Intel(R) Core(TM) i7-12700F 2.10 GHz.

**Table 1.** SVMD and VMD in terms of computational efficiency, memory utilization. Comparative analysis table in terms of usage.

| Method | Time(second) | Memory(MB) | Data Length(day) |
|--------|--------------|------------|------------------|
| VMD | 1.53 | 6.28 | 7 |
| SVMD | 1.02 | 5.69 | 7 |
| VMD | 5.34 | 10.89 | 15 |
| SVMD | 2.49 | 7.09 | 15 |
| VMD | 15.14 | 20.06 | 30 |
| SVMD | 6.64 | 12.97 | 30 |

**Comment 6**

Line 127: Please provide the full forms of GRU and LSTM when first mentioned.

**Response:**

Thanks for your suggestion.

We have used the full form of the GRU and LSTM models first mentioned in line 127 of the original manuscript, modifying them to "Gated Recurrent Units (GRU)", "Long Short-Term Memory (LSTM)". We have also revised line 126 of the original manuscript to read "Fandom Forest (RF)" in order to maintain the uniformity of the article.

**Comment 7**

Lines 134 & 180: Please correct "Mado earthquake" to "Maduo earthquake".

**Response:**

Thanks for your suggestion.

We have replaced "Mado earthquake" with "Maduo earthquake" in lines 134 and 180 of the original manuscript. At the same time, we have corrected the incorrect "Mado earthquake" to the correct "Maduo earthquake" in the rest of the manuscript.

**Comment 8**

(1)Line 135: Why was the Menyuan station selected for detecting precursors of the Maduo earthquake? A significant M6.9 earthquake occurred near Menyuan on January 1, 2022. It is more plausible that the signals from the Menyuan station correspond to the local mainshock rather than to the Maduo earthquake over 400 km away.

(2)Lines 190–195: As noted by the authors, I believe the signals described may be related to the Menyuan earthquake in 2022 rather than the Maduo earthquake in 2021.

**Response:**

Thanks for your suggestion.

We understand your legitimate concerns regarding the selection of the Menyuan station for detecting precursors of the Maduo earthquake. The Menyuan station is about 422.06 km from the epicenter of the Maduo earthquake, and the Menyuan station is about 35.44 km from the epicenter of the Menyuan earthquake. Both the Maduo and Mengyuan earthquakes are within the observation area of the Mengyuan station, and the drilling strain instruments are able to pick up the signals associated with the earthquakes. Therefore, your concern is reasonable.

However, it should be noted that the study of borehole strain data mainly focuses on short-term and proseismic anomaly characteristics (3 months before the earthquake). And many scholars have used borehole strain data to extract short-term and proseismic anomalies before earthquakes (Zhu et al., 2018; Ma and Zhang, 2014; Li et al., 2024a). In this study, the anomalies we extracted from the Menyuan station appeared about 3 and 2 months before the Maduo earthquake, a time scale consistent with typical short-term precursor characteristics. And our results are consistent with the theory of fault synergistic process of (Ma et al., 2014), and we believe that the anomalies observed before the Maduo earthquake are related to the gestation process of the earthquake. The time gap between the Menyuan magnitude 6.9 earthquake, which occurred on January 8, 2022, and our use of the Menyuan station to extract the pre-seismic anomalies of the Maduo earthquake is about 11 months, which is far beyond the time frame of short-term proseismic anomalies. Therefore, we believe that the anomalies extracted from the Mengyuan station this time belong to the pre-seismic precursor of the Maduo earthquake, rather than the Mengyuan earthquake. Therefore, we choose the Menyuan station to detect the precursor of the Maduo earthquake.

In addition, we calculated the monitoring range of the short-term precursor anomaly of this Maduo Ms7.4 based on the empirical equations of magnitude-short-term precursor anomaly time and magnitude-short-term precursor range given by (Su, 1991). The empirical formulas are shown below:

$$logT = 0.49M - 1.25 \tag{1}$$

$$logR^{(0)} = 0.182M + 1.4 \tag{2}$$

where $M$ denotes the magnitude and $R$ denotes the short-term precursor monitoring range of the borehole strain gauge. Based on the above empirical formula, we substitute the magnitude Ms7.4 for the present study, and obtain that the anomaly time of the strain short-term precursor corresponding to an earthquake of magnitude 7.4 is about 237 days (about 7.9 months). The monitoring range of the short-term precursor of the borehole strain gauge is about 559 km. The distance between the Menyuan station and the epicenter in this study is 422.06 km, which indicates that the station we chose has the ability to receive the short-term precursor anomalies of the borehole strain gauge. Meanwhile, the anomaly time of the strain short-term precursor corresponding to the magnitude 7.4 earthquake is about 7.9 months, which indicates that the strain short-term precursor of the Menyuan earthquake is not the pre-seismic anomaly of the Maduo earthquake that we extracted. Therefore, we believe that the anomaly we extracted from the Mengyuan station records the Maduo earthquake, not the Mengyuan earthquake.

Finally, we very much agree with your important point about the correlation of anomalous signals with local earthquakes. In fact, this is one of the special concerns in our study. In order to further verify this conclusion, we will carry out the extraction and analysis of pre-seismic precursor anomalies of the Mengyuan earthquake for the Mengyuan station in our follow-up study.

**Comment 9**

Line 140: When decomposing the borehole strain data, is there a risk that potential precursor signals might be lost or distorted in the process?

**Response:**

Thanks for your suggestion.

According to your proposal, we may have distorted precursor signals when disaggregating the borehole strain data, but did not lose potential precursor signals. In order to verify that we are not losing the potential precursor signals when decomposing the borehole strain data, we randomly selected the raw data of three anomalous days as shown in Fig. 1. It is clear in Fig. 1 that the anomalous days we defined exhibit short-period, high-frequency oscillatory signals in the raw waveforms, suggesting that these days are related to crustal activity. After we processed them using SVMD, we were still able to successfully extract these days as anomalous days. Therefore, it can be shown that our study did not lose the precursor signals when decomposing the borehole strain data. We may have distorted precursor signals when decomposing the borehole strain data. However, we studied the data from the perspective of amplitude and frequency, not from the perspective of data morphology. Therefore, the distortion of anomalous morphology will not affect the results of this study.

[Figure]

**Figure 1.** Plot of raw data for three randomly selected anomalous days.

**Comment 10**

Line 230: You use a 7-day sliding window with a 1-day step for anomaly detection. Please justify this choice and provide statistical or empirical evidence supporting the selected window size and step.

**Response:**

Thanks for your suggestion.

In the SVMD method, our criteria for selecting the sliding window size are based on the equipment capacity and the efficiency of data processing to start with, and the optimal window size is selected through experiments. Our preprocessed dataset is the borehole strain data recorded from January 1, 2020 to May 31, 2021 at the Mengyuan station. As shown in Table 2, we chose sliding window sizes of 7, 15 and 30 days, respectively, and the time and memory size needed for the calculation process are given in Table 2. The correlation between the data cannot be maintained if the sliding window is chosen too small. Whereas, a window that is too large significantly increases the memory consumption and computation time, resulting in the program not being able to be executed. Considering the time spent on the SVMD calculation process and the memory size of the computer. Therefore, we choose a parameter configuration with a window length of 7 days and a sliding step of 1 day. Running device name: DESKTOP-FPCL3PM; Running device version: Windows 11 Professional; Running device processor: 12th Gen Intel(R) Core(TM) i7-12700F 2.10 GHz.

**Table 2.** The experimental results of SVMD correspond to different sliding window sizes.

| Window(day) | Time(second) | Memory(MB) |
|---|---|---|
| **7Days** | 1.12 | 5.90 |
| **15Days** | 2.39 | 6.89 |
| **30Days** | 7.24 | 13.68 |

**Comment 11**

Line 334. The study uses strain data from January 2020 to May 2021. However, this

relatively short time span may not sufficiently capture long-term pre-seismic cycles. Please discuss the temporal adequacy of the dataset.

**Response:**

Thanks for your suggestion.

In this study, we choose the borehole strain data from January 2020 to May 2021 at Menyuan station because the study of borehole strain data mainly focuses on short-term and near-seismic anomalies. Meanwhile, in order to show that our study focuses on short-term and near-seismic anomalies, we add the phrase "For short-term and proseismic anomaly extraction from borehole strain data." in line 333 of the original manuscript.

In addition, it can be observed from Fig. 9 of the original manuscript that, compared with other anomalies, the anomalous range coverage of our extracted borehole strain data is more comprehensive, and at the same time, it can better reflect the anomalous characteristics of the short-term pre-earthquake and near-earthquake phases. Therefore, it shows that our selection of the borehole strain data from January 2020 to May 2021 for the study has a certain degree of temporal appropriateness.

[Figure]

**Figure 9 of the original manuscript. Summary of our study of borehole strain anomalies (light blue) versus subsurface-to-atmosphere anomalies (from light green to gray) for the Maduo Ms7.4 earthquake. The red dashed line indicates the date of the earthquake.**

**Comment 12**

Line 340: Key model parameters (e.g., SVMD bandwidth = 2000, convergence threshold = 1e-7, Informer layers = 128) are presented without any sensitivity analysis. The authors should evaluate how parameter changes affect signal decomposition quality (e.g., signal-to-noise ratio). Additionally, please include uncertainty bounds in prediction intervals (e.g., in Figure 7) to better reflect model confidence and variability.

**Response:**

Thanks for your suggestion.

(1) Line 340: Key model parameters (e.g., SVMD bandwidth = 2000, convergence

threshold = 1e-7, Informer layers = 128) are presented without any sensitivity analysis. The authors should evaluate how parameter changes affect signal decomposition quality (e.g., signal-to-noise ratio).

The noise reduction function of SVMD is realized by applying Wiener filtering to remove Gaussian noise for each mode during the decomposition process. The parameter $\alpha$ controls the bandwidth of the Wiener filter, when the value of parameter $\alpha$ is larger, the Wiener filter will remove more noise, however, too large a value of parameter $\alpha$ will affect the convergence of the algorithm. On the contrary, if a smaller $\alpha$ is chosen, the algorithm will converge more easily, but more noise will remain (Wu et al., 2018). In SVMD decomposition, the value of the parameter $\alpha$ affects the accuracy of the decomposition and the length of the decomposition time, and too high a value of $\alpha$ can cause the program to enter a dead loop (Priyanka et al., 2015). The parameter $\alpha$ can be chosen manually based on numerical experiments, and in this paper, 2000 is chosen as the optimal value. Reducing the convergence tolerance parameter can make the algorithm converge to more accurate results, but it will also increase the computation time. Therefore,for comprehensive consideration we choose 1e-7.

In order to analyze the bandwidth of the appropriate SVMD decomposition, we choose one month's data from the Mengyuan station for the decomposition. The decomposition results are shown in Fig. 3, Fig. 4, and Fig. 5.

[Figure]

**Fig.3. Decomposition results of SVMD with bandwidth equal to 100.**

[Figure]

**Fig.4. Decomposition results of SVMD with bandwidth equal to 1000.**

[Figure]

**Fig.5. Decomposition results of SVMD with bandwidth equal to 2000.**

From Figs. 3, 4, and 5 we can observe that in Figs. 3 and 4, when the bandwidth is small, the k1 layer contains significant periodic variations, resulting in a failure to separate the trend term from the solid tides. Therefore, we cannot choose a value with a small bandwidth. And when the bandwidth is large, it causes the program to enter a dead loop. Therefore, by manual selection, this study chooses the bandwidth equal to 2000 as the optimal value,as shown in Fig. 5.

In addition, for the selection of the dimensionality of the SVMD-informer model, we reduced the model fit by decreasing the number of layers of the model. As shown in Figs. 7 and 8, when the dimensionality of the model is too high, the SVMD-informer model outputs obvious overfitting between the predicted values and the real values, which leads to the inability to effectively extract the anomalous precursors before the earthquake. When the model dimension is too low, the model learns fewer features, resulting in a higher deviation between the predicted and real values, and the overall prediction performance decreases. We finally chose the number of model layers as 128 through a large number of experiments. As shown in Fig. 6, a model dimension of 128 layers can effectively reduce the fit between the predicted values and the true values output from the SVMD-informer model, and can successfully extract the pre-seismic anomaly precursors. In addition, moderately reducing the model dimension can also reduce the consumption of computational resources and improve the overall experimental efficiency.

[Figure]

**Figure 6. informer model with 128 layers.**

[Figure]

**Figure 7. informer model with 256 layers.**

[Figure]

**Figure 8. informer model with 512 layers.**

(2) Additionally, please include uncertainty bounds in prediction intervals (e.g., in Figure 7) to better reflect model confidence and variability.

Regarding your question "please include uncertainty bounds in prediction intervals", we have carefully considered and supplemented our analysis. Our SVMD-Informer model is determined by the upper and lower bounds of the prediction sequence output by the decoder, using the normal distribution method. The prediction interval of the network consists of the upper and lower bounds, and this prediction interval serves the same purpose as the confidence interval you mentioned. As you say, Figure 7 in the original manuscript does not indeed convey the reliability and variability of the results in a completely clear way, so we have supplemented it with statistics on daily anomaly rates. We did a count of the eligible judgment results for each day and used the statistical results as the number of anomalies for each day, and calculated the daily anomaly rate based on the number of anomalies for each day, as shown in Figure 9.

[Figure]

Fig. 9. Statistical results of daily anomaly rate at Mengyuan station. (a) Statistical results of daily anomaly rate from November 1, 2020 to May 31, 2021 . (b) Results of daily anomaly rate statistics from February 1, 2021 to May 22, 2021.

As shown in Figure 9(a), only a small number of phenomena meeting the criteria for anomalous days occurred between 7 months before the earthquake and 3 months before the earthquake, while from 3 months before the earthquake until the earthquake occurred, there was a significant increase in the number of anomalous phenomena. Figure 9(b) highlights the detailed changes from 3 months before the earthquake to the occurrence of the earthquake. We find that the daily anomaly rate increases from mid-February to mid-March. Similarly, the occurrence daily anomaly rate increases rapidly from the end of March to the time of the earthquake. These results corroborate with the

S-fit results in Fig. 8 of the original manuscript and provide a more complete picture of the process of anomaly changes before the earthquake. Thus, it can better reflect the confidence and variability of our model.

[Figure]

**Figure 8 of the original manuscript. Cumulative results of anomalous days of borehole data at Mengyuan station (MY). The red dashed line indicates the date of the earthquake, and the blue and red curves indicate the results of the S-fit function for the first and second phases, respectively.**

**Comment 13**

Line 340: The patterns shown appear more indicative of the coseismic effects of the Maduo earthquake, rather than its precursors. Please consider adding information in Figure 6 to clarify and support the presence of pre-seismic signals.

**Response:**

Thanks for your suggestion.

In response to your comments, and in order to clarify and support the existence of a pre-seismic signal, we have added more detailed information to Figure 6 of the original manuscript. As shown in Figure 6 of the original manuscript, k1 indicates the trend, k2 indicates the solid tide, and k3+k4+k5 indicate the strain associated with the earthquake. From the figure, we can clearly see that the anomalies in the borehole strain data at the Menyuan station before the earthquake are more likely to be precursor signals of the Maduo earthquake rather than co-seismic effects of the Maduo earthquake.

[Figure]

**Figure 6 of the original manuscript. Data for** $S_a, S_{13}, S_{24}$ **at the Mengyuan station and the results of the SVMD decomposition of** $S_a$**. k1 denotes the trend term, k2 denotes the earth tides, and k3, k4, and k5 denote the strains associated with the earthquakes. The red dashed line indicates the date of the earthquake.**

**Comment 14**

Line 380: The two-stage S-shaped anomaly accumulation (Figure 8) is interesting, but no underlying mechanism is provided. Moreover, the upward trend of pre-seismic acceleration could plausibly be associated with the Menyuan earthquake in January 2022 instead of the Maduo event.

**Response:**

Thanks for your suggestion.

In response to your suggestion, we have revised lines 400-404 of the original manuscript to better clarify the mechanism behind the two-stage S-type anomaly accumulation by revising it to read, "In addition, the two-stage accelerated growth of the borehole strain accumulation results may reveal two preparatory mechanisms prior to the mainshock. This is consistent with the theory of fault synergistic process of (Ma et al., 2014). They found that the occurrence of earthquakes is closely related to the three-stage synergistic evolution of faults through an indoor experimental study of plane-walk-slip faults. In the first stage, the initial stress nonlinear divergence leads to localized weakening and the formation of discrete strain release zones. The second stage is characterized by an increase in stress and a widening of the strain release zone. In the final stage, the expansion of the strain-release region and the rapid increase in the strain level in the strain-accumulation region. The anomalous cumulative acceleration about 3 months before the Maduo earthquake corresponds to the first and second stages in the theory, which is manifested by the deviation of the stress curve from linearity and the beginning of the formation and slow expansion of the discrete release zone. The acceleration 2 months before the earthquake reflects the characteristic changes of the third stage, which is characterized by the accelerated expansion of the release zone and the sharp increase of the strain in the accumulation zone. Therefore, we believe that the anomalies observed before the Maduo earthquake are related to the

process of earthquake incubation.". By modifying this paragraph we explain in detail the mechanism behind the two-stage S-type anomaly accumulation. Therefore, we conclude that the accelerated rise of the two-stage S-type anomaly accumulation in this study is related to the Maduo earthquake rather than the Mengyuan earthquake.

**Comment 15**
Line 415: The study references various anomalies (CO, TBB, electron density), but does not sufficiently address the possibility of false positives caused by anthropogenic or environmental factors. Please discuss the limitations of borehole strain data, including potential sensitivity to non-tectonic influences such as groundwater fluctuations or temperature changes.
**Response:**
    Thanks for your suggestion.
(1) Line 415: The study references various anomalies (CO, TBB, electron density), but does not sufficiently address the possibility of false positives caused by anthropogenic or environmental factors.
Based on your comments, we will discuss whether external influences may have an effect on other anomalies. (Shi et al., 2024) showed that the epicenter of the Maduo earthquake was far away from the industrial and densely populated areas, the CO was little influenced by human factors, and the atmospheric background concentration of CO was more stable, which could well reflect the characteristics of the local environmental changes, and it was a good case to study the relationship between seismic activities and gas anomalies. Meanwhile, they analyzed the April CO data from 2014-2022 at the epicenter. The results showed that the CO concentration in April 2021 was significantly higher than the levels in the same period of other years, and the peak occurred on April 30th. Similar anomalies were only observed during the same period in 2015 and 2016, which corresponded to the previous Ms4.1 earthquake. This suggests that the CO anomaly in 2021 is more likely to originate from seismic activity rather than seasonal factors. (Yang et al., 2024) systematically excluded environmental interference such as meteorological factors, diurnal variations and earth rotation by combining wavelet transform and Fourier transform to ensure the specificity of TBB anomaly signals. Specifically, the db8 wavelet base is used to filter out high and low frequency interferences and retain the effective signal in the middle frequency. Meanwhile, effects such as solar radiation are further circumvented by night-time data selection and power spectrum analysis, thus effectively distinguishing seismic-related anomalies from environmental noise. In addition, (Tian et al., 2023) used the Improved Pattern Informatics (IPI) method to effectively reduce the interference of anthropogenic and environmental factors through multiple technical means. Firstly, an adaptive background trend removal algorithm is utilized to eliminate systematic biases caused by long-term space weather activities such as solar radiation and geomagnetic storms, while space weather parameters such as the Kp index and the Dst index are combined for simultaneous screening in order to exclude anomalous data during magnetic disturbances. Finally, the influence of environmental noise such as short-term atmospheric gravity waves is suppressed by statistical normalization of sliding time

windows, and continuous anomalies with seismic correlation are effectively distinguished from transient disturbances with the help of cross-validation of diurnal and nocturnal orbital data. Therefore, all of the above researchers have excluded the influence of anthropogenic or environmental factors on the anomalies they extracted.

(2)Please discuss the limitations of borehole strain data, including potential sensitivity to non-tectonic influences such as groundwater fluctuations or temperature changes.

Borehole strain monitoring is a technical means of recording rock or crustal deformation by installing strain sensors deep underground. This monitoring method can accurately capture the crustal microstrain induced by plate tectonic movement or seismic activity, and provide an important basis for studying the dynamics of the regional tectonic stress field. Strain data are usually particularly sensitive to changes in geostress caused by fault activity, and can provide valuable information for the extraction of earthquake precursors and the study of seismicity mechanisms.

At the same time, we fully agree that borehole strain data, while highly sensitive to tectonic deformation, can also be affected by non-tectonic environmental factors or human interference. Borehole strain gauges are susceptible to factors such as instrument drift, solid tide, temperature, barometric pressure and rainfall. Without adequate preprocessing, they may be mistaken for tectonic strain signals. In the data preprocessing, we adopted the Segmented Variational Modal Decomposition (SVMD) method, which effectively strips out low-frequency components such as annual cycle and solid tide. Meanwhile, I added the analysis for temperature, barometric pressure and rainfall data for the study time range, as shown in Fig. 10.

[Figure]

Figure 10. (a) Three-hourly regional variations of barometric pressure, temperature and precipitation in the Mengyuan area during the period from January 1, 2020 to May 31, 2021. (b) Differential processing results of three-hourly regional variations of barometric pressure, air temperature and precipitation in Mengyuan area during the period from January 1, 2020 to May 31, 2021.

In Fig. 10(a), we analyze the three-hourly regional variations of barometric pressure, temperature, and rainfall in the Mengyuan area (35.97 to 39.97°N, 99.4 to 103.4°E)

from NASA's Giovanni-4 platform (https://giovanni.gsfc.nasa.gov/giovanni, last accessed May 9, 2025) , with a time frame of January 1, 2020 to December 31, 2021. The results show that the barometric pressure and air temperature fluctuate inversely within a certain range, while the rainfall gradually decreases year by year after peaking in summer, reflecting a significant annual cycle. In addition, in order to minimize the influence of external factors on the borehole strain data, we carried out differential processing on the three-hourly regional averages of barometric pressure, air temperature and rainfall in Mengyuan area. Differential processing was utilized to remove the effect of cyclic variations, thereby highlighting anomalies in the data. The results of the processing are shown in Fig. 10(b), which shows that there are no anomalous changes in the three-hourly regional averages of barometric pressure and air temperature and rainfall in Mengyuan. We exclude the influence of temperature, barometric pressure and rainfall on the anomalies observed in the pre-earthquake borehole data from Maduo. Therefore, we have reason to believe that the anomalies we extracted before the Maduo earthquake are related to the earthquake genesis process. Also add in line 426 of the original manuscript " Although borehole strain monitoring techniques are capable of accurately capturing crustal microstrain induced by plate tectonic movements or seismic activity, they are susceptible to factors such as temperature, air pressure, and rainfall. For this purpose we analyzed regional three-hourly variations of barometric pressure, air temperature and rainfall in the Mengyuan area (35.97 to 39.97°N, 99.4 to 103.4°E) from NASA's Giovanni-4 platform (https://giovanni.gsfc.nasa.gov/giovanni, last accessed May 9, 2025) , with a time frame of January 1, 2020 to December 31, 2021. In Fig. 10(a), the barometric pressure and air temperature fluctuate inversely within a certain range, while the rainfall gradually decreases year by year after peaking in summer, reflecting a significant annual cyclicity. In addition, in order to minimize the influence of external factors on the borehole strain data, we performed differential processing on the three-hourly regional averages of barometric pressure, air temperature, and rainfall in the Mengyuan area. Differential processing was utilized to remove the effect of cyclic variations, thereby highlighting anomalies in the data. The results of the processing are shown in Fig. 10(b), which shows that the three-hourly regional means of barometric pressure, air temperature, and rainfall in Mengyuan do not show any anomalous changes. We exclude the influence of barometric pressure, air temperature and rainfall on the anomalies observed in the pre-earthquake borehole data from Maduo. Therefore, we have reason to believe that the anomalies we extracted before the Maduo earthquake are related to the earthquake genesis process.".

**References**

Chen, Y. and Lü, P.: Analysis of Coseismic Response of PBO Borehole Strainmeters, Geodesy and Geodynamics, 2019.

Chi, C., Li, C., Han, Y., Yu, Z., Li, X., and Zhang, D.: Pre-earthquake anomaly extraction from borehole strain data based on machine learning, Scientific Reports, 13, 10.1038/s41598-023-47387-z, 2023.

Cui, Y., Huang, J., Zeng, Z., and Zou, Z.: CO Emissions Associated with Three Major Earthquakes Occurring in Diverse Tectonic Environments, Remote Sensing, 16, 10.3390/rs16030480, 2024.

Guo, J., Li, W., Liu, X., Wang, J., Chang, X., and Zhao, C.: On TEC anomalies as precursor before MW 8.6 Sumatra earthquake and MW 6.7 Mexico earthquake on April 11, 2012, Geosciences Journal, 19, 721-730, 10.1007/s12303-015-0005-6, 2015.

Han, C., Yan, R., Marchetti, D., Pu, W., Zhima, Z., Liu, D., Xu, S., Lu, H., and Zhou, N.: Study on Electron Density Anomalies Possibly Related to Earthquakes Based on CSES Observations, Remote Sensing, 15, 10.3390/rs15133354, 2023.

Li, C., Qin, C., Zhang, J., Duan, Y., and Chi, C.: Analysis of Borehole Strain Anomalies Before the 2017 Jiuzhaigou Ms7.0 Earthquake Based on Graph Neural Network, EGUsphere, 10.5194/egusphere-2024-2025, 2024a.

Li, C., Duan, Y., Han, Y., Yu, Z., Chi, C., and Zhang, D.: Extraction of pre-earthquake anomalies from borehole strain data using Graph WaveNet: a case study of the 2013 Lushan earthquake in China, Solid Earth, 15, 877-893, 10.5194/se-15-877-2024, 2024b.

Li, F., Zhang, B., Verma, S., and Marfurt, K. J.: Seismic signal denoising using thresholded variational mode decomposition, Exploration Geophysics, 49, 450-461, 10.1071/eg17004, 2018.

Li, M., Yao, L., Wang, Y., Parrot, M., Hayakawa, M., Lu, J., Tan, H., and Xie, T.: Anomalous phenomena in DC–ULF geomagnetic daily variation registered three days before the 12 May 2008 Wenchuan $M_S$ 8.0 earthquake, Earth and Planetary Physics, 3, 330-341, 10.26464/epp2019034, 2019.

Liu, J., Cui, J., Zhang, Y., Zhu, J., Huang, Y., Wang, L., and Shen, X.: Study of the OLR Anomalies before the 2023 Turkey M7.8 Earthquake, Remote Sensing, 15, 10.3390/rs15215078, 2023.

Liu, Q., Zhang, J., Chi, S., and Yan, W.: Time-frequency characteristics of four-component borehole strain at Guzan Station before and after the 2013 Lushan Ms7.0 earthquake, Journal of Seismology, 2014.

Ma, J., Guo, Y., and Sherman, S. I.: Accelerated synergism along a fault: A possible indicator for an impending major earthquake, Geodynamics & Tectonophysics, 5, 387-399, 10.5800/gt-2014-5-2-0134, 2014.

Ma, Z. and Zhang, X.: Analysis on borehole strain anomaly before the strong earthquakes, Earthquake Research, 2014.

Ouyang, Z.: U.S. PBO Project:Borehole strainmeter networks face a challenge, International Seismological Dynamics, 10.3969/j.issn.0235-4975.2011.10.005, 2011.

Priyanka, D., Udit, S., and Barathram, R.: Single Channel Blind Source Separation Based on Variational Mode Decomposition and PCA, IEEE, 2015.

Qi, Y., Wu, L., Ding, Y., Liu, Y., Chen, S., Wang, X., and Mao, W.: Extraction and Discrimination of MBT Anomalies Possibly Associated with the Mw 7.3 Maduo (Qinghai, China) Earthquake on 21 May 2021, Remote Sensing, 13, 10.3390/rs13224726, 2021.

Qiu, Z., Tang, L., Zhang, B., and Guo, Y.: In situ calibration of and algorithm for strain monitoring using four-gauge borehole strainmeters (FGBS), Journal of Geophysical Research: Solid Earth, 118, 1609-1618, 10.1002/jgrb.50112, 2013.

Rao, D., Huang, M., Shi, X., Yu, Z., and He, Z.: A Microseismic Signal Denoising Algorithm Combining VMD and Wavelet Threshold Denoising Optimized by BWOA, Computer Modeling in Engineering & Sciences, 141, 187-217, 10.32604/cmes.2024.051402, 2024.

Shi, Y., Xin, C., Liang, H., and Liu, H.: CO anomalies before and after the 2021 Maduo Ms7.4 earthquake in Qinghai Province, Journal of Earthquake Engineering, 10.20000/j.1000-0844.20230105002, 2024.

Su, K.: EARTHQUAKE-MONITORING CAPABILITY OF BOREHOLE STRAINMETER, EARTHQUAKE, 1991.

Thomas, H., Yun-Tai, C., Paolo, G., Raul, M., Ian, M., Warner, M., Gerassimos, P., Gennady, S., Koshun, Y., and Jochen, Z.: OPERATIONAL EARTHQUAKE FORECASTING. State of Knowledge and Guidelines for Utilization, Annals of Geophysics, 54, 10.4401/ag-5350, 2011.

Tian, W., Zhang, Y., Ju, C., Zhang, S., Feng, M., and Liu, F.: An Improved Pattern Informatics Method for Extracting Ionospheric Disturbances Related to Seismicity Based on CSES Data: A Case Study of the Mw 7.3 Maduo Earthquake, Authorea Preprints, 10.22541/essoar.170365237.70486329/v1, 2023.

Wu, W., Wang, Z., Zhang, J., Ma, W., and Wang, J.: Research of the Method of Determining $k$ Value in VMD based on Kurtosis, Mechanical Drives, 2018.

Xue, Y.-j., Cao, J.-x., Wang, X.-j., Li, Y.-x., and Du, J.: Recent Developments in Local Wave Decomposition Methods for Understanding Seismic Data: Application to Seismic Interpretation, Surveys in Geophysics, 40, 1185-1210, 10.1007/s10712-019-09568-2, 2019.

Yang, M., Zhang, X., Zhong, M., Guo, Y., Qian, G., Liu, J., Yuan, C., Li, Z., Wang, S., Zhai, L., Li, T., and Shen, X.: Spatio–Temporal Evolution of Electric Field, Magnetic Field and Thermal Infrared Remote Sensing Associated with the 2021 Mw7.3 Maduo Earthquake in China, Atmosphere, 15, 10.3390/atmos15070770, 2024.

Yi, Z.: Short term and imminent abnormality for borehole volume strain observation data, Seismic and Geomagnetic Observation and Research, 2005.

Zhu, K., Chi, C., Yu, Z., Fan, M., Li, K., and Sun, H.: The characteristics analysis of strain variation associated with Wenchuan earthquake using principal component analysis, Annals of Geophysics, 63, 10.4401/ag-7946, 2020.

Zhu, K., Chi, C., Yu, Z., Zhang, W., Fan, M., Li, K., and Zhang, Q.: Extracting borehole strain precursors associated with the Lushan earthquake through principal component analysis, Annals of Geophysics, 61, 10.4401/ag-7633, 2018.

---

## Author Response (AR2)

**Response to Reviewer:**

I greatly appreciate your valuable comments on the manuscript. Thank you for your insightful suggestions. All of your feedback is highly significant and has provided important guidance for both our paper and our research. We have revised the manuscript in accordance with your suggestions. Below is our detailed response to each revision:

**Comment 1**

Cite the reference of Eq 3.

**Response:**

Thanks for your suggestion.

We have added a reference to the literature cited by Dobrovolsky in the original manuscript at line 174 to ensure the accuracy of the citation. A revised manuscript with the correction marked in red at Line 173 (Page 6) has been attached as the supplemental material entitled "revised manuscript".

**Comment 2**

Section 3 is very large; authors should describe a summary of SVMD and Informer network methods. In my opinion such detailed description of the methods it is not necessary. For a better lecture such methods can be summarize in order to focus to the analysis instead of the details of both methods.

**Response:**

Thanks for your suggestion.

We have refined the content and optimized the structure of Section 3, SVMD-Informer. To improve the clarity of the explanation, we omitted the detailed description of the parameters in Equation (5) in Section 3.1 (SVMD), retaining only an overview of the variational model and augmented Lagrangian used in SVMD.

In Section 3.2 on the Informer network, we first removed the lengthy explanation of Informer features to reduce the number of paragraphs in the paper. Second, we condensed the original manuscript lines 261–266 and integrated the revised content with the paragraph starting from line 267 to ensure a more coherent narrative. Additionally, the content from lines 279 to 281 of the original manuscript has been summarized and integrated with the relevant content from lines 255 to 256 to avoid repetitive descriptions. Finally, the two paragraphs from lines 306 to 318 of the original manuscript have been merged to further reduce the number of paragraphs and enhance the overall conciseness of the expression.

Following these revisions, the content of Section 3.2 on the Informer network is now clearer, providing an overview of the Informer network's background and its applications in other fields, introducing the Informer network, followed by a general description of its key attention mechanism, encoder, and decoder, and providing the parameters of the model trained in this paper and the upper and lower bounds of data construction during model output. Therefore, after revision, Section 3 on SVMD-Informer is now more concise and focused, with the key points highlighted. A revised manuscript with the corrections marked in red at Line 238 (Page 10), Lines

240–241 (Page 10), Lines 246–248 (Page 10), and Lines 290–291 (Page 12) has been attached as the supplemental material entitled "revised manuscript".

**Comment 3**

The phrase "where the query vector represents the current network output state", in line 263 is repeated.

**Response:**

   Thanks for your suggestion.

In Comment 2, we have already modified line 263 of the original manuscript. Therefore, the revised manuscript avoids the problem of repeated phrases.

**Comment 4**

Regarding the training and validation (line 324) authors should explain the criteria to chose that period.

**Response:**

   Thanks for your suggestion.

This paper selects data from the first eight months of 2020 as the training set and validation set because the data during this period is relatively stable, with few abnormal disturbances, which can effectively optimize model training, thereby improving prediction performance and increasing the accuracy of anomaly extraction. At the same time, in order to maintain the continuity of the data in the time dimension and avoid interference with model training due to a large time span, data from this relatively stable period is selected as the training set and validation set. A revised manuscript with the correction marked in red at Line 301-303 (Page 12) has been attached as the supplemental material entitled "revised manuscript".

**Comment 5**

Line 331 must be corrected the expression "1e - 7".

**Response:**

   Thanks for your suggestion.

We have corrected line 331 of the original manuscript from "1e - 7" to $10^{-7}$, in accordance with the standard format for scientific notation. A revised manuscript with the correction marked in red at Line 308 (Page 13) has been attached as the supplemental material entitled "revised manuscript".

**Comment 6**

In Section 5 Line 343 authors wrote "The anomalies were recognized when the raw data exceeded the corresponding upper or lower bounds.". Where such bounds are defined?

**Response:**

   Thanks for your suggestion.

The definition of "upper and lower bounds" in this paper is determined based on the normal distribution method, as specifically defined in (13) of the original manuscript. The bounds are computed using the following formulas:

$$Lower=Predicition-Z{\times}mad,$$
$$Upper=Predicition+Z{\times}mad,$$

where *Prediction* is the predicted value output by the decoder, Z is the Z score under a normal distribution, and *mad* represents the median absolute deviation. Based on this formula, we can identify outliers for each day. Subsequently, according to the criteria for defining abnormal days in lines 351 to 354 of the original document, we extract the precursors to earthquakes.

**Comment 7**

In Figure 8 there are points from 2020/11/01 until 2021/01/30. Is there some interpretation of such period?

**Response:**

Thanks for your suggestion.

In the original manuscript Figure 8, although it includes some data points from November 1, 2020, to January 30, 2021, our analysis found that there were only a few abnormal days between November 1 and December 31, 2020, and the overall data trend from November 1, 2020, to January 30, 2021, was relatively stable, without showing a significant surge trend. Therefore, we do not consider the period from November 1, 2020, to January 30, 2021, to be a distinct phase of abnormal accumulation.

[Figure]

**Original manuscript Figure 8. Cumulative results of anomalous days of borehole data at Menyuan station (MY). The red dashed line indicates the date of the earthquake, and the blue and red curves indicate the results of the S-fit function for the first and second phases, respectively.**

**Comment 8**

In the meteorological parameters showed in Figure 10 It is not clearly distinguish the anomalies described in the section. Can the authors identify such anomalous behavior?

**Response:**

Thanks for your suggestion.

To better illustrate that the anomalies we extracted are not influenced by meteorological factors, we marked the anomalous time periods of the borehole strain data in Figure 10 of the original manuscript (two green dashed lines indicate the first phase of anomalies, and two royal blue dashed lines indicate the second phase of anomalies). In Figure 10(a) of the original manuscript, atmospheric pressure and temperature exhibit synchronous fluctuations within a certain range, while precipitation reaches its peak annually during summer and then gradually declines, showing a significant annual cyclical pattern. However, there is no obvious correspondence between this cyclical pattern and the anomalies discussed in this paper. Additionally, to eliminate the influence of cyclical variations, we applied differential processing to the three-hour meteorological factors in the region. As shown in Figure 10(b) of the original manuscript, the processed results still do not show a clear correspondence with the anomalies discussed in this paper. Therefore, we conclude that the extracted anomalies are unrelated to meteorological factors and are more likely associated with the earthquake incubation process. Finally, we have provided supplementary explanations in the original manuscript. A revised manuscript with the corrections marked in red at Lines 404–406 (Page 17), Lines 409–412 (Page 17), and Lines 418–419 (Page 18) has been attached as the supplemental material entitled "revised manuscript".

[Figure]

**Original manuscript Figure 10. (a) Three-hourly regional variations of barometric pressure, temperature and precipitation in the Menyuan area during the period from January 1, 2020 to May 31, 2021. (b) Differential processing results of three-hourly regional variations of barometric pressure, air temperature and precipitation in Menyuan area during the period from January 1, 2020 to May 31, 2021. The red dashed line indicates the date of the earthquake, the two green dashed lines indicate the first phase anomaly, and the two royal blue dashed lines indicate the second phase anomaly.**

**Comment 9**

The conclusions must be improved.

**Response:**

Thanks for your suggestion.

We have optimized the conclusion section. Specifically, Lines 442–453 of the original manuscript have been revised to: "The method addresses the issues of slow computation speed and memory limitations in traditional VMD by adopting SVMD, and significantly improves the accuracy and stability of long-sequence time series prediction by integrating the Informer network. By analyzing the borehole strain data from the Menyuan station, we successfully identified two distinct phases of anomalous cumulative acceleration preceding the Maduo earthquake, occurring approximately three and two months before the event, respectively. The cumulative anomaly curves exhibited a characteristic S-shaped growth pattern. This finding is consistent with the fault synergy process theory proposed by(Ma et al., 2014), further supporting the correlation between borehole strain anomalies and the Maduo earthquake. With the continued progress of machine learning technology and the ongoing accumulation of seismic observation data, this method is expected to provide higher-precision technical support for earthquake prediction and help reduce seismic disaster risk.". A revised manuscript with the corrections marked in red at Lines 422–431 (Pages 18–19) has been attached as the supplemental material entitled "revised manuscript".

**References**

Ma, J., Guo, Y., and Sherman, S. I.: Accelerated synergism along a fault: A possible indicator for an impending major earthquake, Geodynamics & Tectonophysics, 5, 387-399, 10.5800/gt-2014-5-2-0134, 2014.

**Other red markings in the revised manuscript explain**

**Line 248 (Page 10), Lines 251-254 (Pages 10-11), Line 257 (Page 11), Line 273 (Page 11), Line 292-293 (Pages 12).**

Correct modification of the formula.

**Line 270 (Page 11).**

Correct order of images.

**Lines 294-295 (Page 12).**

Modify to italics corresponding to the formula.

**Line 331 (Page 14).**

The word "fulfill" here modifies "meet".

**Line 353 (Page 15).**

The word "leveled" here modifies "levelled".

**Lines 465-466 (Page 19).**

Correct reference format.

**Lines 472-473 (Page 20).**

Correct reference format.

**Lines 505-507 (Page 20).**

Correct reference format.

**Line 534 (Page 21).**

Correct reference format.

**Lines 227 (Page 9), Lines 567-568 (Page 21).**

Correct reference format.

**All pages.**

All instances of "mengyuan" have been corrected to "menyuan" throughout the revised manuscript.